# Interpretable Estimation of CNN Deep Feature Density using Copula and the Generalized Characteristic Function

## Abstract

We present a novel empirical approach toward estimating the Probability Density Function (PDF) of the deep features of Convolutional Neural Networks (CNNs). Estimating the PDF of deep CNN features is an important task, because it will yield new insight into deep representations. Moreover, characterizing the statistical behavior has implications for the feasibility of promising downstream tasks such as density based anomaly detection. Expressive, yet interpretable estimation of the deep feature PDF is challenging due to the Curse of Dimensionality (CoD) as well as our limited ability to comprehend high-dimensional interdependencies. Our novel estimation technique combines copula analysis with the Method of Orthogonal Moments (MOM), in order to directly estimate the Generalized Characteristic Function (GCF) of the multivariate deep feature PDF. We find that the one-dimensional marginals of non-negative deep CNN features after major blocks are not well approximated by a Gaussian distribution, and that the features of deep layers are much better approximated by the Exponential, Gamma, and/or Weibull distributions. Furthermore, we observe that deep features become increasingly long-tailed with network depth, although surprisingly the rate of this increase is much slower than theoretical estimates. Finally, we observe that many deep features exhibit strong dependence (either correlation or anti-correlation) with other extremely strong detections, even if these features are independent within typical ranges. We elaborate on these findings in our discussion, where we hypothesize that the long-tail of large valued features corresponds to the strongest computer vision detections of semantic targets, which would imply that these large-valued features are not outliers but rather an important detection signal.

## 1 Introduction

Convolutional Neural Networks (CNN) have revolutionized the performance of image analysis tasks including image classification, semantic segmentation, object detection, and image and video synthesis (Yuan & Zhang, 2016; Goodfellow et al., 2020; Hao et al., 2020; Xing et al., 2023). At the time of writing, CNNs continue to play a dominant role as image feature encoders for state-of-the-art techniques including several prominent Vision-Language Models (VLMs) (Long, 2024; Radford et al., 2021; Li et al., 2021) as well as diffusion models for image generation (Rombach et al., 2022; Yang et al., 2023). Nevertheless, the extraordinary complexity of CNNs has coined the nick-name of *black-box*, that learns an uninterpretable and high-dimensional feature representation. Although there has been progress toward understanding the semantic meaning of features, there has been much less progress toward understanding the high-dimensional probability distribution that underlies the statistical behavior of deep CNN features (Zhang et al., 2025; Vladimirova et al., 2019; Feldman, 2020; Feldman & Zhang, 2020; Allen-Zhu & Li, 2023; Chen et al., 2024; Giraldo & Schwartz, 2019; Hermann & Lampinen, 2020; Qiu et al., 2024; Shwartz-Ziv & Tishby, 2017).

What is the distribution of deep CNN features? The goal of this work is to improve our understanding of the statistical behavior of the learned representation of deep CNN features through high-dimensional statistical analysis techniques including copula analysis, and the Method of Orthogonal Moments (MOM) (Sklar, 1959; Nelsen, 2006; Yudell, 1975). Rather than assuming a parametric distribution apriori, our approach is to directly estimate the Generalized Characteristic Function (GCF) (Hansen, 1982) of the copula

interdependence term (Nelsen, 2006). Our copula+GCF approach provides a novel and interpretable probability density estimate of CNN features, without rigid parametric assumptions that may be unjustified, and without altering the CNN feature representation which may reduce accuracy. As an empirical technique, we can gain greater insight by plotting and analyzing the marginal and interdependence components of the feature copula density PDF. We want to estimate and observe the probability density of deep features for popular CNN architectures, as we believe that this will lay the groundwork for future feature-density analysis methods that can identify stronger parametric assumptions, thereby leading to improved understanding of the feature space and its statistical behavior.

Several prior works have assumed the CNN features follow a Multi-Variate Gaussian (MVG) (Majurski et al., 2024; Zhu et al., 2022; Lee et al., 2018; Rippel et al., 2021; Zhu et al., 2022). But the motivation for MVG traces back to a handful of histogram plots of the penultimate critic features in the supplemental materials of Lee et al. (2018). There has never been a verification that MVG is suitable to the deeper intermediate features (non-penultimate), which is especially important because in most architectures, deep features undergo ReLU activation which deactivates (zeros out) any negative valued features, thereby enforcing non-negativity. Max-pooling moreover retains the strongest positive features, throwing away weaker less-positive detections. Thus it is important to look at and measure the distribution of non-negative features as these are the only features that have an impact on subsequent activations. We observe that these non-negative deep features are highly non-Gaussian, and increasingly approximate long-tailed distributions with increasing network depth. We furthermore show that at adequate network depth, many deep features exhibit non-linear dependence, with typical feature values showing statistical independence for typical (non-extreme) values, but strong statistical dependence of larger extreme values, a phenomenon known as upper tail dependence (Nelsen, 2006).

Several works have attempted to construct deep generative architectures with explicit feature representations that enable exact or approximate PDF estimates (Kingma, 2013; Caterini et al., 2021; Kingma et al., 2016; Rezende & Mohamed, 2015). Such techniques include Variational Auto-Encoders (VAE) (Kingma, 2013), as well as autoregressive and normalizing flows (Caterini et al., 2021; Kingma et al., 2016; Rezende & Mohamed, 2015). VAE provides a very coarse estimate of the PDF of a deep embedding layer through the incorporation of approximate KL-divergence constraints with an assumed parametric prior (typically standard Gaussian) (Kingma, 2013). Unlike VAE, our approach provides a fine grained estimate of the PDF for multiple feature layers simultaneously and without rigid parametric constraints. Normalizing flows take a very different approach of propagating feature probability density layer-by-layer over a series of reversible layers by means of the change of variables formula (Kingma et al., 2016; Rezende & Mohamed, 2015). Like our technique, normalizing flows provide a fine grained PDF for multiple layers simultaneously, however, our approach has two advantages: 1. The need for *reversible* layers means that normalizing flows cannot be applied to popular CNN architectures such as ResNet and VGG without substantial redesign. 2. Our approach is *interpretable* in the sense that we can make visually understandable plots describing the shape of the feature marginal and interdependence terms. As such, our goal is not only to model the probability density and achieve a good fit, but moreover to represent the probabilities in a way that can facilitate visual exploration of the statistical behavior.

It has been proposed that estimating the probability density of deep CNN features could have many practical downstream applications including Out-of-Distribution (OOD) detection (Lee et al., 2018; Liu et al., 2020; Zhu et al., 2022; Jiang et al., 2023; Le Lan & Dinh, 2021), adversarial detection (Lee et al., 2018; Le Lan & Dinh, 2021), domain generalization (Chen et al., 2024), and federated learning (Sun et al., 2023). It has been anticipated that if one knows the density distribution of deep features, that it would be possible to use outlier detection to statistically distinguish legitimate inlier data from anomalous outlier data, thereby providing a greater level of model robustness to data that is out of model scope (Guérin et al., 2023; Lee et al., 2018; Liu et al., 2020). Perhaps the most straightforward of these downstream tasks is anomaly detection (including OOD detection), which was first proposed by Bishop (1993) as an application of neural network feature density. Density based anomaly detection (outside of deep learning) is a traditional approach with historical success (Chandola et al., 2009). Recently, density based anomaly detectors have been implemented and report good performance on evaluation tasks (Cook et al., 2024; Liu et al., 2022).

But other recent papers cast doubt on whether density based anomaly detection is even possible using deep CNN features (Nalisnick et al., 2018; Lee et al., 2018; Liu et al., 2020; Zhu et al., 2022; Le Lan & Dinh,

2021; Jiang et al., 2023; Zhang et al., 2021). Nalisnick et al. (2018) was the first to observe the strange result that feature density methods, including VAE and normalizing flows, often attribute greater density to the out-of-distribution samples relative to the in-distribution samples. The research community is still searching for an answer as to why traditional density based anomaly detection methods have fallen short when applied to deep features, and several competing hypotheses have been proposed. Hypotheses include the often unaccounted for effects of reparameterization (Le Lan & Dinh, 2021) and class-imbalance (Jiang et al., 2023), as well as questions regarding the empirical validity of the underlying 'typical set' hypothesis (Zhang et al., 2021) stating that there is a discernible difference between the statistical properties of *typical valued* in-distribution and out-of-distribution features (Lee et al., 2018; Liu et al., 2020; Zhu et al., 2022). Although the true reason for this behavior may be difficult to pinpoint, we believe that understanding the statistical behavior of the distribution of deep features plays a vital role in answering this question. If deep CNN features followed the same distribution as traditional anomaly detection datasets, one would expect traditional density-based outlier detection methods to perform well just as they have in the past (Bishop, 1993; Chandola et al., 2009). Empirical work to visually interpret and understand the distribution of deep features will provide new insight toward answering this important question. Based on our results, we propose a new *Long-tailed view* hypothesis, that many deep features exhibit long-tailed marginals for which outliers are likely to occur, and moreover that these outliers correspond to strong computer vision detections of a given target view. Thus, attempting to remove outliers for the purpose of anomaly detection may in fact mistakenly remove the most important detection signal.

There are several reasons to suspect that the marginal distribution of deep CNN features might exhibit a long-tailed distribution, although this phenomenon has not been previously verified through direct analysis of the feature marginals (Zhang et al., 2025; Vladimirova et al., 2019; Feldman, 2020; Feldman & Zhang, 2020). Long-tailed theory suggests that the real-world image set exhibits many *rare* instances, be it classification subcategories, or image statistics. Early works have established that the frequency of object sub-categories naturally follows a long tail (Salakhutdinov et al., 2011; Zhu et al., 2014). It is now understood that a hidden long tail of *rare* instances persists even within class balanced datasets; i.e. imbalanced subcategories exist even if they are not explicitly labeled as such (Feldman, 2020). Could it be that deep CNNs learn long-tailed deep feature marginals in order to better represent these long-tailed image statistics? At present, there is indirect empirical supporting evidence for this hypothesis. Feldman & Zhang (2020) provides rigorous empirical evidence that popular CNNs are *memorizing rare* instances, in order to more accurately model *long-tailed* image statistics hidden within popular image datasets. Feldman & Zhang (2020) uses a cross-validation approach to identify the long tail of *rare* instances within the dataset. The CNN performance significantly drops if these *rare* instances are excluded from training. Moreover, simply fine-tuning the last layer with these *rare* instances is not adequate to restore model performance. Therefore it is concluded that the memorization of the long-tail of *rare* instances occurs within the deeper convolution layers. But we wonder: Could the deeper convolution layers exhibit long-tailed feature marginals in order to better represent the long-tail of *rare* instances? Our empirical analysis of marginals shows that for all of the models and datasets under consideration, that at least one such deeper layer exhibits long-tailed feature marginals.

The strongest theoretical evidence in support of long-tailed deep CNN feature marginals comes from the analysis of (Vladimirova et al., 2019). (Vladimirova et al., 2019) proves that Bayesian neural networks (including Bayesian CNNs), exhibit long tailed deep features for Layer 3 and deeper under the assumption of independent unit Gaussian inputs and weight priors. More-specifically, these deep features are sub-Weibull with optimal Weibull tail parameter $\theta = \frac{1}{2}i$ where $i$ is the depth of the CNN layer. Optimal tail parameter $\theta$ is considered to be the largest value of $\theta$ for which the marginal is sub-Weibull. This means conv Layer 1 is sub-Gaussian ($\theta = \frac{1}{2}$), Layer 2 is sub-exponential ($\theta = 1$), and subsequent layers are long-tailed (Layer 3: $\theta = 1\frac{1}{2}$, Layer 4: $\theta = 2$, Layer 5: $\theta = 2\frac{1}{2}$ etc.). This theoretical result very clearly shows how long-tailed deep feature marginals can arise from a neural network even in the absence of long-tailed image statistics, as the assumed Gaussian inputs are certainly not long-tailed. Nevertheless, the proof requires strong assumptions on the distribution of weights and inputs, it is not known to what extent this proof may be useful in explaining the empirical distribution of deep feature marginals for popular CNN architectures and image datasets. In our section on Analysis of marginals, we numerically estimate the optimal Weibull $\theta$ parameters for ResNet and VGG architectures in several image datasets, and show that the empirical behavior of $\theta$ is largely inconsistent with the predicted results of Vladimirova et al. (2019), although there are some similarities.

Notably, the interior layers of the network do indeed show some steady increase in $\theta$ parameter with layer depth consistently across all models and datasets, although the rate of increase is significantly slower than the theoretical $\theta = \frac{1}{2}i$. Moreover, we find that for all models and architectures, at least one sufficiently deep intermediate layer is indeed long-tailed $\theta > 1$, but depending on the model and architecture, many layers may also be sub-exponential $\theta < 1$. Finally, the optimal $\theta$ value for the deepest conv layer appears to depend heavily on the dataset, with more complex image datasets (CIFAR-100, Imagenette2) exhibiting larger $\theta$ values than simpler image datasets (CIFAR-10, MNIST) suggesting that image statistics also play a role in whether the deepest layers are long-tailed.

A contribution of our work is that we are the first to apply copula analysis toward understanding the statistical behavior of deep CNN features. Moreover, we believe that we are the first to apply copula analysis toward answering any basic research question in the area of deep machine learning. Copula analysis is a popular method for understanding multivariate joint probabilities in domains such as economics (Kim, 2020), physical sciences (Bhatti & Do, 2019), and healthcare (Demongeot et al., 2013). Bhatti & Do (2019) refers to copulas as *"the most powerful tools that can model dependent structures between various complex correlated variables."* Indeed, copula have the ability to not only model the joint density, but also yield visually expressive plots to facilitate greater understanding of the statistical behavior. Nevertheless, the vast majority of works that have attempted to combine copula with deep learning have been highly applied toward the same traditional problem domains for which copula analysis was historically popular (Huang et al., 2022; Ouyang et al., 2019; Bedoui et al., 2023; Silva et al., 2020). The method of Ling et al. (2020) is notable as it provides a deep learning approach toward estimating the parameters of an Archimedean copula. However, even the benchmarks within Ling et al. (2020) are highly applied toward the traditional financial modeling and econometrics domains. We are unaware of any prior work that applies copulas toward better understanding of the behavior of deep learning models. Our primary goal is to gain empirical insight into the question: *what is the probability distribution of deep CNN features?*.

Another contribution of our approach is that our novel copula model exhibits greatly improved expressive power because we are the first to compute a GCF for the purpose of representing the copula density (Nelsen, 2006; Yudell, 1975; Hansen, 1982). This is a large improvement over standard Archimedean copulas which are inflexible, because the joint copula density depends on only on a single rank correlation coefficient, typically either Spearman's $\rho$ or Kendall's $\tau$ (Nelsen, 2006). For Archimedean copulas, one must choose from a parametric *generating* function such as the Gumbel, Frank, or Clayton copula generators. The choice of the parametric generator gives some blunt flexibility over the presence of upper and lower tail dependence in the bivariate copula (Nelsen, 2006). This makes Archimedean copulas quick to compute and easy to interpret, but the rigid parametric assumptions raise the risk of assuming an unjustified joint distribution, especially if one does not perform extensive exploratory analysis beforehand. By comparison, our GCF copula also includes Spearman's $\rho$, as this fully defines to the orthogonal moment $\mu_{1,1}$ corresponding of the bivariate linear Legendre polynomial term. But in addition to Spearman $\rho$, we also take into account a much larger number of rank-shape statistics as part of the GCF, as indeed all of the moments constitute different rank-shape statistics. This allows us to non-parametrically model the copula density rather than assuming a simple parametric form. Moreover, the GCF easily generalizes to multivariate joint densities, whereas multivariate Archimedean copulas are typically implemented using either a series of bivariate joint copulas, or by assuming only a single rank-order statistic for the entire multivariate copula (Nelsen, 2006). As such, our proposed copula is far more expressive than the standard Archimedean copula design, and is therefore able to more accurately model the joint probability density, while maintaining the ease of visualization, and incorporating the familiar Spearman's $\rho$ term within the copula density calculation.

In recent years, orthogonal functions and the method of moments have found a variety of uses in deep feature spaces. These techniques can allow for compression of the feature space, as well as have led to the development of improved loss functions for GANs and domain adaptation. PCA decomposition of deep feature covariane is a viable approach toward model compression while maintaining predictive power (Garg et al., 2019), orthogonal gradient updates can also preserve prior knowledge and prevent catastrophic forgetting (Saha et al., 2021; Farajtabar et al., 2020). PCA is however a purely linear method. The method of moments has also found utility as a way to introduce non-linearity within orthogonal deep feature decompositions (Majurski et al., 2024; Li et al., 2015; 2017; Ansari et al., 2020). Notably, the Maximum Mean Discrepancy

(MMD) statistic is widely used as a loss function for deep generative models. MMD is in fact mathematically derived from the method of moments by using the kernel trick with Radial Basis Functions (RBF) (Li et al., 2015; 2017). Ansari et al. (2020) introduces a Characteristic Function Distance (CFD) based on the Fourier moment series as a fast alternative to MMD loss for generative models. Majurski et al. (2024) introduces a method of moments loss function based on the power series which is used as embedding constraint for contrastive learning. Although the method of moments has found utility within deep representation learning, it is almost always used only as a *loss* function. In theory, the method of moments is also capable of recovering the full probability density of the deep features, but has rarely been applied for this purpose outside of niche applications.

Characteristic functions are in fact a powerful and expressive way of estimating probability density, although this approach has found limited and niche applications (Hansen, 1982; Yu et al., 2023; Choi et al., 2020; Nolan, 2013). The Empirical Characteristic Function (ECF) was first introduced by Feuerverger & Mureika (1977) who advocated its use for many problems. McCulloch (1996) Argues that financial modeling and time series forecasting would greatly benefit from the ECF estimation. Yu (2004) shows that ECF outperforms Gaussian Mixture Models (GMM) in modeling financial timeseries forecasts. Nolan (2013) further analyzes the properties of the ECF and provides numerical routines for estimation of densities as applied to timeseries data. Yu et al. (2023) combines ECF with probabilistic circuits (Choi et al., 2020) to construct a powerful hybrid density estimation technique as estimated using UCI benchmark datasets, and found this approach to be competitive with state of the art density estimation techniques. Although ECF are capable of density estimation, their application has been slow to gain traction outside of the financial modeling, and timeseries forecasting community. Moreover, we are unaware of prior works that have incorporated ECF as a means to compute copula density as we have within our methodology.

Our copula+GCF approach also exhibits improved expressive power over ECF in modeling distributions with long-tailed and/or discontinuous marginal densities. The traditional ECF attempts to approximate a probability distribution by estimating its Fourier transform. But the Fourier transform, which is based on *sin* and *cos* terms may require an impractical number of frequencies in order to approximate functions with power law asymptotes (as in long-tailed marginals) and/or discontinuities as are prevalent in non-negative distributions. There are also open questions about how best to sample from the ECF. By separating the marginal and interdependence terms, our copula+GCF strategy allows for empirical characteristic approximation of multivariate distributions that would be impractical to sample from using a standard ECF approach. Furthermore, our GCF construction provides a straightforward sampling strategy as it is based on a discreet well ordered series of orthogonal functions, rather than a continuous set of orthogonal functions as in the ECF. As such, we find that our copula+GCF approach performs favorably to the ECF when used to estimate the density of deep CNN features.

Our method is designed to both expressively model the *probability density* of deep features while also making it easier for a human to visually *interpret* the *statistical behavior* of features through copula analysis and visualization of the marginal and interdependence terms. As such, by *interpretability* we are referring to the ability to construct the probability density function in a way that leads to interpretation of the *sample statistics*. This is distinct from the more common notion of interpretability in machine learning, including network architectures with the ability to highlight the semantic meaning of features, and/or provide visual explanations of predictions (Noh et al., 2015; Kolouri et al., 2017; Zhang et al., 2018; Allen-Zhu & Li, 2023; Chen et al., 2024; Hermann & Lampinen, 2020). High dimensional statistics are notoriously difficult to comprehend by a human, and copula analysis can help with this interpretation. As such, our empirical method provides an expressive ability to reconstruct the probability density of deep features, while also providing a new visual aid for researchers who may be interested in understanding the statistical distribution of feature marginals and interdependence.

## 2 Methodology

### 2.1 Copula Analysis

In high-dimensional statistics, copula analysis is a powerful method that allows one to completely separate the marginal distribution of the random variables from their interdependence. Given a set of random variables $(X_1, X_2, \ldots, X_D)$ and marginal cumulative distribution functions $(F_1, F_2, \ldots, F_D)$, one can perform the following probability integral transform.

$$(Y_1, \ Y_2, \ \ldots \ , \ Y_D) \quad = \quad (F_1(X_1), \ F_2(X_2), \ \ldots \ , F_D(X_D)) \tag{1}$$

The copula $C$ is defined as the cumulative distribution function of the probability integral transform of the random variables, as given by the following.

$$C(y_1, \ y_2, \ \ldots \ , \ y_D) \quad = \quad Pr\,[Y_1 \le y_1, \ Y_2 \le y_2, \ \ldots \ , \ Y_D \le y_D] \tag{2}$$

Moreover the copula density is the probability density function $c(y_1, \ y_2, \ \ldots \ , y_D)$ associated with cumulative copula distribution $C(y_1, \ y_2, \ \ldots \ , y_D)$, which is defined by the following.

$$c(y) = c(y_1, \ y_2, \ \ldots \ , y_D) = \frac{\partial \ C(y_1, \ y_2, \ \ldots \ , y_D)}{\partial y_1 \ \partial y_2 \ \ldots \ \partial y_D} \tag{3}$$

Typically the probability integral transform converts a marginal distribution into a uniform distribution on the interval $(0, 1)$. However, in our approach, we carry out our analysis using a re-scaled version of the probability integral transform to a uniform the interval $(-1, 1)$. This is because many well-known orthogonal functions are designed for this interval, and these orthogonal functions allow us to measure the copula density term in greater detail without parametric assumptions. This modified probability integral transform is defined by the following.

$$F_d(x) = 2 \ Pr[X_d \le x] - 1 \tag{4}$$

### 2.2 Method of Orthogonal Moments (MOM)

The method of moments are a way of fully describing the shape of a probability distribution through the use of *consistent estimators*, which asymptotically share sample and population statistics. Assume that $x$ is a finite sample of $N$ elements drawn from an infinite population, then the following shows a series of well-behaved sample statistics $\hat{\mu}_t$ should very closely approximate their population statistics $\mu_t$.

$$\hat{\mu}_t \approx \mu_t \quad \text{where} \quad \mu_t = E(\phi_t(x)) \quad \text{and} \quad \hat{\mu}_t = \frac{1}{N} \sum_{i=1}^{N} \phi_t(x_i) \tag{5}$$

The original method of moments simply defines basis functions as $\phi_t$ as the power functions $\phi_t(x) = x^t$ for $t = 1, 2, 3, 4 \ldots$. In this case, the zero-mean samples would correspond to the *mean, variance, skewness, kurtosis*, etc. But these basis functions have the disadvantage of being non-orthogonal, and thereby duplicating some shape information. It is more powerful to choose $\phi_t$ to represent an orthogonal basis set, such as the Fourier series $\phi_t = e^{i\pi t x}$, or one of the orthogonal polynomial sets such as the Legendre or Chebyshev polynomials thereby representing the Method of Orthogonal Moments (MOM).

We use the MOM to analyze the copula interdependence $Y_1, \ \ldots \ , Y_D$ of deep features separately from the marginals $f(X_1), \ \ldots \ , f(X_D)$. As such, we must estimate the interdependence between multiple random variables by defining multivariate moments in terms of the expected product of univariate basis statistics in the copula space. If we define $T$ as a $D$ dimensional vector of integers where $T_d$ represent the desired

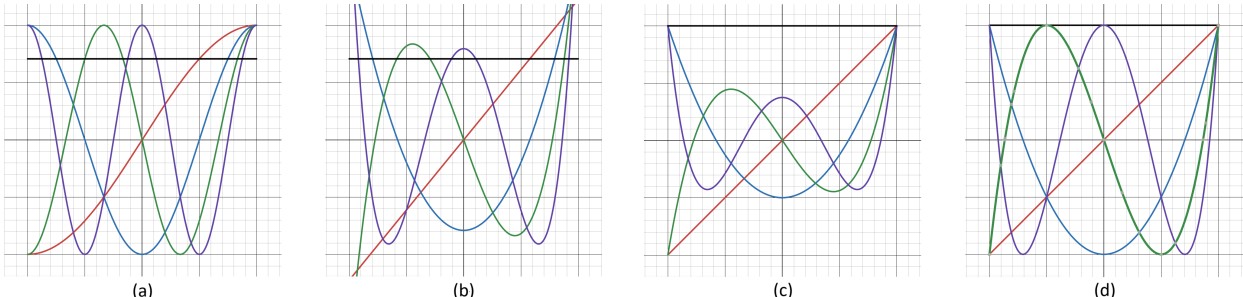

Figure 1: Comparison of orthogonal basis functions over the uniform interval $(-1, 1)$. (a) Real-valued Fourier series (b) Normalized Legendre polynomials (c) Legendre polynomials without normalization (d) Chebyshev polynomials.

moment of the $d^{th}$ random variable $Y_d$, then the joint moment $\mu_T$ corresponds to inner product of basis vectors given by the following.

$$\mu_T = E\left(\prod_{d=1}^{D} \phi_{T_d}(Y_d)\right) \qquad \hat{\mu}_T = \frac{1}{N}\sum_{i=1}^{N}\left(\prod_{d=1}^{D} \phi_{T_d}(Y_{d,i})\right) \tag{6}$$

For our analysis, we specifically define $\phi_t$ as either the real-valued Fourier series or the normalized Legendre polynomial series because in addition to being orthogonal over the rescaled copula interval of $(-1, 1)$, these basis functions also have the useful property that all non-constant basis terms exhibit zero integral over $(-1, 1)$, which simplifies our Generalized Characteristic Independence (GCI) metric. The specific forms of these series that we propose also exhibit the property of having unit $L_2$ norm over the target interval $(-1, 1)$. The Legendre polynomials also have the advantage that, like the power series, *mean* and *covariance* are part of the basic shape descriptors, which are highly familiar concepts thereby aiding in practical interpretation.

### 2.3 Generalized Characteristic Function

A useful property of the MOM, is that it allows one to recover the actual copula probability density function $c(y)$ completely non-parametrically, and without any overly-rigid assumptions on the shape of this distribution by means of the Generalized Characteristic Function (GCF). The original Characteristic Function refers to the observation that if one defines the basis set as the Fourier series, then the following shows how the population moments resembles the Fourier transform of the PDF.

$$\mu_p = E\left(\phi_p(y)\right) = \int_{-\infty}^{\infty} c(y)\, e^{i\pi ty} dy \tag{7}$$

As such, one can recover the copula density by means of an inverse discrete Fourier transform of the population moments $\mu$. If this process is performed using the sample moments $\hat{\mu}$ then using the following, one recovers the sample estimate of the copula density function $\hat{c}$. In the univariate case, this process of recovering the copula density from the characteristic function can be written as follows.

$$c(y) = \sum_{t=1}^{K} \mu_t\, \phi_t(y) \qquad \hat{c}(y) = \sum_{t=1}^{K} \hat{\mu}_t\, \phi_t(y) \tag{8}$$

The GCF refers to the straightforward extension of this technique to non-Fourier orthogonal moments. Analogously, one can recover the full copula density using a discrete inverse Fourier-like transform for any orthogonal basis set. Nevertheless, it is highly desirable to select basis functions that exhibit the following properties,

- Orthogonal over unit interval $(-1, 1)$

- Real valued, exhibiting even and odd harmonics

- Unit length $L_2$ norm over interval $(-1, 1)$

- All non-constant moments exhibit zero integral over $(-1, 1)$

In order to adhere to these properties, we propose to employ a specific normalized form of the Legendre polynomials, as well as a real-valued form of the Fourier series as we describe in further detail.

## 2.4 Normalized Legendre Polynomials

The Legendre Polynomials (figure 1b,c) are a set of real-valued orthogonal basis functions over the target interval $(-1, 1)$ with several desirable properties. Unlike the Chebyshev polynomials (figure 1d), the Legendre polynomials (figure 1c) exhibit zero integral over the interval $(-1, 1)$, except trivially for the constant polynomial $P_0$. This zero-integral property is highly-desirable for our resultant GCF. The following polynomials can be generated efficiently using Bonnet's recurrence.

$$
\begin{aligned}
P_0(y) &= 1 \\
P_1(y) &= y \\
P_{n+1}(y) &= \frac{2n+1}{n+1} y P_n(y) - \frac{n}{n+1} P_{n-1}(y)
\end{aligned}
\tag{9}
$$

The Legendre polynomials in this form do not exhibit unit-length $L_2$ norm over the interval $(-1, 1)$, as such, we propose normalizing the Legendre polynomials based on their $L_2$ norm in order to obtain unit-length orthogonal moments as in (figure 1b). This normalized form is obtained from the following.

$$
\phi_t(y) = \frac{P_t(y)}{||P_t||_2} \quad \text{where} \quad ||P_t||_2 = \sqrt{\int_{-1}^{1} P_t^2(y) \, dy}
\tag{10}
$$

### 2.4.1 Real-valued Fourier Series

As our sample is real-valued, one can equivalently represent the Fourier series as a sum of real-valued *cos* (even) and *sin* (odd) harmonic terms. Moreover, it is possible to simplify this to only *cos* terms if one makes use of the trigonometric phase identity, which is given by the following.

$$
sin(y) = cos(y - \frac{\pi}{2} + 2\pi t) \qquad \text{for integer } t
\tag{11}
$$

As such, one can present both even and odd real-valued Fourier basis functions using an elegant and simplified form, demonstrated by the following.

$$
\begin{aligned}
\phi_0(y) &= \frac{\sqrt{2}}{2} \\
\phi_t(y) &= cos\left(t\frac{\pi}{2}(y-1)\right)
\end{aligned}
\tag{12}
$$

The real-valued Fourier basis functions in this form are shown in (figure 1a). One can see that these basis functions correspond analogously one-to-one with the Legendre and Chebyshev polynomials, with $\phi_t$ exhibiting exactly $t$ roots over the target interval $(-1, 1)$. This form of the Fourier series also exhibits our desired properties, as the non-constant basis functions have zero integral over the target interval $(-1, 1)$. Moreover, for all basis functions the $L_2$ norm over the target interval is exactly equal to 1 when presented in this form.

## 2.5 Multivariate Empirical GCF Estimate of Copula Density

We now define our empirical estimate of the multivariate copula density $\hat{c}(\vec{y})$. Thus far we have discussed an empirical estimate of the copula density in the univariate case $\hat{c}(y)$ as based on the Legendre and real-value Fourier series. Note that $T$ is a $D$-dimensional multivariate counter of radix K, i.e. $T \in \mathbb{Z}_K^D$. The $d^{th}$ element of this counter, $T_d$ indicates the degree of the orthogonal function used for the $d^{th}$ rank-ordered feature $Y_d$. Based on this, we can define the multivariate orthogonal functions $\Phi_T(\hat{y})$ as the following.

$$\Phi_T(\vec{y}) = \prod_{d=1}^{D} \phi_{T_d}(\vec{y}_d) \tag{13}$$

As such, the multivariate empirical copula density can be obtained by the inner product of multivariate moments $\hat{\mu}$ and the corresponding orthogonal functions $\Phi$.

$$\hat{c}(\vec{y}) \quad = \quad \hat{\mu} \; \cdot \; \Phi(\vec{y}) \quad = \sum_{T \, \in \, \mathbb{Z}_K^D} \hat{\mu}_T \; \Phi_T(\vec{y}) \tag{14}$$

## 2.6 Generalized Characteristic Distance and Independence

Given a set of orthogonal population moments $\mu_T$ and $\nu_T$ for probability distributions $P$ and $Q$ respectively, the Generalized Characteristic Distance (GCD) fully describes the difference in shape between probability distribution as the Manhattan distance of the Fourier-like transforms of the PDFs. The following shows how this is calculated by taking the Manhattan distance between the moments.

$$D_{char}(P, Q) = \sum_{T \, \in \, \mathbb{Z}_K^D} |\mu_T - \nu_T| \tag{15}$$

In the event that a set of features are completely independent, then the copula density $c(y_1, \ldots y_D)$ corresponds to the uniform distribution on the hypercube $y_i \in (-1, 1)$. As such, we new define a Generalized Characteristic Interdependence (GCI) metric $H_{char}(c)$ as the GCD between the copula distribution $c$ and the ideal uniform copula density $Q_{unif}$. $H_{char}$ is exactly zero when variables $y_1, \ldots, y_D$ are statistically independent, and nonzero when these variables show some statistical dependence along one or more orthogonal moments. $H_{char}(c)$ is defined by the following.

$$H_{char}(c) = D_{char}(c, Q_{unif}) \tag{16}$$

The real-valued Fourier and normalized Legendre series have the convenient property that all basis functions integrate to zero over $(-1, 1)$ (except trivially the constant basis function $\phi_0$). As such, it is straightforward to show that the ideal uniform distribution $Q_{unif}$ has all zero moments. Therefore, for the Fourier and Legendre moments, the GCI simplifies to L1 norm, as shown in the following.

$$H_{char}(c) = ||\mu||_1 \qquad \text{for Fourier and Legendre moments} \tag{17}$$

## 2.7 Algorithmic Design

We now describe our algorithm for calculation of a multivariate copula density of deep features based on our methodology. Algorithms 1 and 2 show the calculation of marginal and interdependence terms. Algorithm 3 further shows the estimation of the copula density. Both algorithms depend on additional pseudocode for evaluation of orthogonal functions, as well as a multivariate counter as provided in appendix E.

---

**Algorithm 1** Calculation of copula marginal terms

---

1: **procedure** CALCULATEMARGIANLS(X,D,N)
2:     **for** $d \leftarrow 1 \ldots D$ **do**
3:         ▷ *Calculate Probability Integral Transform*
4:         **for** $i \leftarrow 1 \ldots N$ **do**
5:             $Y_{d,i} \leftarrow 2Pr[X_d \leq X_{d,i}] - 1$
6:         **end for**
7:     **end for**
8:     **return** $(X, Y)$
9: **end procedure**

---

**Algorithm 2** Calculation of copula interdependence terms

---

1: **procedure** CALCULATEINTERDEPENDENCE(Y,D,N,K)
2:     ▷ *Initialize empirical GCF Density*
3:     $T = $ INITIALIZE()
4:     **while** $T \neq$ overflow **do**
5:         $\hat{\mu}_T \leftarrow 0$
6:         $T \leftarrow$ INCREMENT($T$)
7:     **end while**
8:
9:     ▷ *Calculate empirical Multivariate Orthogonal Moments*
10:     **for** $i \leftarrow 1 \ldots N$ **do**
11:
12:         ▷ *Univariate orthogonal functions evaluated at sample $Y_i$*
13:         $\phi \leftarrow$ EVALORTHOGONAL($Y_{1:D, i}$)
14:
15:         ▷ *Multivariate orthogonal moments $\hat{\mu}$*
16:         $T = $ INITIALIZE()
17:         **while** $T \neq$ overflow **do**
18:             **if** ORDERK($T$) **then**
19:                 $$\hat{\mu}_T \leftarrow \hat{\mu}_T + \frac{1}{N} \prod_{d=1}^{D} \phi_{d,T_d}$$
20:             **end if**
21:             $T \leftarrow$ INCREMENT($T$)
22:         **end while**
23:
24:     **end for**
25:     **return** $\hat{\mu}$
26: **end procedure**

---

---

**Algorithm 3** Calculation of copula density for $y$

---

1: **procedure** CopulaDensity($\vec{y}, \hat{\mu}$)
2:     $\hat{c} \leftarrow 0$
3:
4:     ▷ *Evaluate univariate functions at test sample y*
5:     $\phi \leftarrow$ EvalOrthogonal($\vec{y}$)
6:
7:     ▷ *Multivariate copula density $\hat{c}$*
8:     $T =$ Initialize()
9:     **while** $T \neq$ overflow **do**
10:         $\Phi_T \leftarrow \prod_{d=1}^{D} \phi_{d,T_d}$
11:         $\hat{c} \leftarrow \hat{c} + \hat{\mu}_T \ \Phi_T$
12:         $T \leftarrow$ Increment($T$)
13:     **end while**
14:     **return** $\hat{c}$
15: **end procedure**

---

## 2.8 Formal Definition

Define $X \in \mathbb{R}^{D \times N}$ as the raw sample under consideration for copula analysis. In our case, $X \in \mathbb{R}^{D \times N}$ represents the activation of different $D$ features over a sample of $N$ instances from the training dataset. As such $X_d \in \mathbb{R}^N$ represents the training sample for the $d^{th}$ feature under consideration.

$$X \in \mathbb{R}^{D \times N} \text{ is the raw sample of } D \text{ dimensions and } N \text{ instances} \tag{18}$$

Define $\mathcal{G}_d$ as the marginal distribution for sample $X_d$.

$$X_d = (X_{d,1}, X_{d,2}, \ldots, X_{d,N}) \text{ is drawn from marginal } \mathcal{G}_d \tag{19}$$

Define $F_d(x) : \mathbb{R} \to \mathbb{R}$ as a probability integral transform that rescales values from the marginal distribution of sample $X_d$ into a uniform distribution over the interval $(-1, 1)$.

$$F_d(x) = 2 \, Pr[X_d \leq x] - 1 \tag{20}$$

Define $Y \in \mathbb{R}^{D \times N}$ as the sample as transformed by the probability integral transform. As such, $\forall d \in [1, D] \quad \forall i \in [1, N]$,

$$Y_{d,i} = F_d\left(X_{d,i}\right). \tag{21}$$

Therefore, $\forall d \in [1, D]$, $Y_d$ follows a uniform distribution over the target interval $(-1, 1)$.

$$Y_d = (Y_{d,1}, Y_{d,2}, \ldots, Y_{d,N}) \text{ is drawn from } Uniform\,(-1, 1) \tag{22}$$

Define $\phi_t(y) : \mathbb{R} \to \mathbb{R}$ as the $t^{th}$ orthogonal function as previously defined in either equation 12 for the *Real Fourier* series or 10 for the *Normalized Legendre* series.

$$\phi_t(y) : \mathbb{R} \to \mathbb{R} \text{ is the } t^{th} \text{ orthogonal function from a series of } K \text{ or more functions} \tag{23}$$

Notice that we consider only the first $K$ functions of the orthogonal series. As such, $t$ belongs to the rank $K$ modular integers, i.e. $t \in \mathbb{Z}_K$ because $t$ can only take the values $\{0, 1, ..., K-1\}$.

$$t \in \mathbb{Z}_K \text{ is a univariate index of the orthogonal function} \tag{24}$$

Define $T \in \mathbb{Z}_K^D$ as a multivariate counter of rank $D$ and modulo $K$. Thus, $T$ allows us to uniquely index each of the multivariate orthogonal functions and moments.

$$T = (T_1, T_2, \dots T_D) \text{ is a multivariate counter of rank } D \text{ and modulo } K \tag{25}$$

Define $\Phi_T(\vec{y}) : \mathbb{R}^D \to \mathbb{R}$ the multivariate orthogonal function of index $T$, which is the product of the orthogonal univariate orthogonal functions.

$$\Phi_T(\vec{y}) = \prod_{d=1}^{D} \phi_{T_d}(\vec{y}_d) \tag{26}$$

Define $\mu_T$ as an unobserved multivariate population moment, and $\hat{\mu}_T$ as the corresponding multivariate sample moment.

$$\mu_T = \mathbb{E}\left(\Phi_T(Y)\right) \qquad \hat{\mu}_T = \frac{1}{N} \sum_{i=1}^{N} \left[ \prod_{d=1}^{D} \phi_{T_d}(Y_{d,i}) \right] \tag{27}$$

Define $\hat{c}(\vec{y}) : \mathbb{R}^D \to \mathbb{R}$ as the empirical copula density as evaluated over transformed testing feature vector $\vec{y}$. $c(\vec{y})$ can be calculated by performing the inverse discreet transform from the orthogonal series. Given that our orthogonal series requires the properities of orthogonality over the unit interval $(-1, 1)$, as well as unit length $L_2$ norm over this interval, the inverse is simplifies to the transpose. Therefore the empirical copula density $\hat{c}(\vec{y})$ is recovered by taking the inner product of the set of multivariate moments $\hat{mu}$ with the set of orthogonal functions $\Phi$ evaluated at vector $\vec{y}$.

$$\hat{c}(\vec{y}) \quad = \quad \hat{\mu} \, \cdot \, \Phi(\vec{y}) \quad = \sum_{T \, \in \, \mathbb{Z}_K^D} \hat{\mu}_T \, \Phi_T(\vec{y}) \tag{28}$$

## 3 Experimental Setup

For our experiments we evaluate three CNN architectures ResNet-18, ResNet-50 , and VGG-19 across four image classification datasets MNIST, CIFAR-10, CIFAR-100, and Imagenette2 (Deng, 2012; Krizhevsky et al., 2009; Deng et al., 2009; Howard, 2019; He et al., 2016; Simonyan & Zisserman, 2014). Imagenette2 is a subset of ImageNet exhibiting 10 classes with the full resolution images that are center-cropped to $224 \times 224$ pixels.

Figure 2 shows the extracted feature spaces from the ResNet and VGG architectures. For each image, after every major convolutional block, we obtain a tensor of size $[N_{img}, \text{filters}, \text{rows}, \text{cols}]$. Due to the self-similarity of features in each row and column, we evaluate the distribution for each of the filters over the entire training or test set. Thus we have a sample of $N = N_{img} \times \text{rows} \times \text{cols}$ for every filter for each block.

As we perform copula analysis, we separately analyze the marginal and the copula interdependence terms. This means that we have separate results and separate analysis for the univariate marginal density, versus the multivariate interdependence. For the marginals, we evaluate 1D probability density functions for each feature. For the copula we analyze the independence of pairs and groups of features. Our exploratory plots of copula interdependence are evaluated for pairs of features. Nevertheless, our method is capable of measuring the density of higher dimensional groups of features which we validate by a task of predicting the distribution of groups of test features given the set of training features.

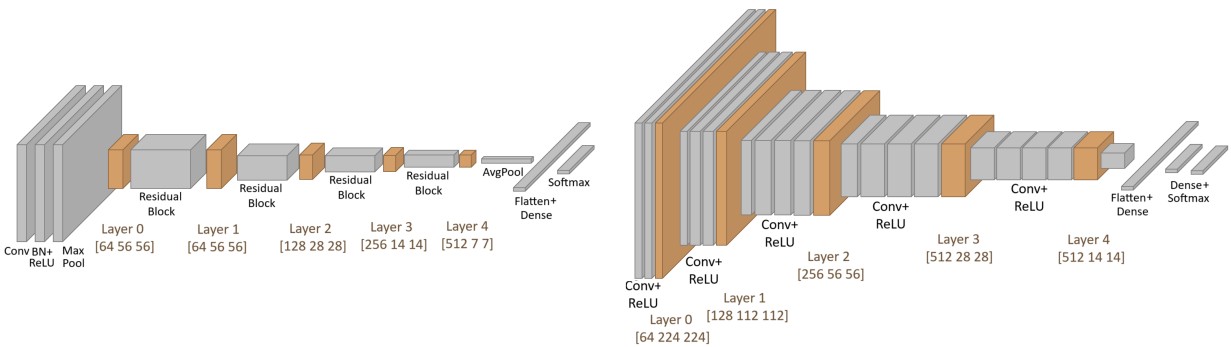

Figure 2: Illustration of ResNet-18 (left) and VGG-19 (right) deep feature layers selected for probability density analysis using copula. Orange shaded regions represent deep feature layers after major architectural blocks that were selected for density analysis. ResNet-50 is not shown, but similar to ResNet-18 (left).

For Imagenette2, we used the pre-trained versions of ResNet-18, ResNet-50, and VGG-19 as included with PyTorch, with a custom trained final linear classification layer. For MNIST, CIFAR-10, and CIFAR-100 the standard versions of ResNet and VGG are not designed to work with such small resolution images, and thus we used the small-image optimized versions of these architectures by Kuang (2017). This small-image optimized version is widely used, often without attribution, in papers that achieve high accuracy on these datasets. The small-image ResNet and VGG architectures were trained from scratch using a learning rate of 0.01, momentum 0.09, and the SGD optimizer.

Rectified Linear (ReLU) is a very common activation function in CNNs, and both ResNet and VGG employ ReLU in order to introduce non-linearity. ReLU also has the additional effect of forcing all-features to be non-negative. As such, the CNN features after major convolutional blocks such as those shown in figure 2 are always non-negative and entirely reside in the positive quadrant of the feature space. For the 1D analysis, we measure the marginal distribution by first measuring the percentage of non-negative features, as well as fitting the observed univariate distribution of non-negative features. The combination of these measurements fully describes the univariate marginal term. For the multivariate copula interdependence term, we add an infinitesimal random jitter to the zero-valued features in order to ensure a statistically independent ordering of zero valued features for the copula interdependence. We present and describe plots of the copula density in two dimensions, and further analyze the goodness of fit of high-dimensional feature copula through a KL-divergence task.

## 4 Results

As we are performing copula analysis, we present separate analysis of the univariate marginal distribution, and of the multivariate copula interdependence terms. The analysis of marginals shows the distribution of feature density including the percentage of non-zero features, the distribution of non-zero features, as well as an analysis of the optimal Weibull tail parameter to determine if the marginals are long-tailed. The copula density interdependence term shows the interdependence of the features with the marginal distribution removed by means of a probability integral transform. The interdependence is modeled by measuring the GCF using the MOM. We present results showing qualitative description of the feature distribution as well as quantitative goodness of fit using KL-divergence and/or cross entropy loss.

### 4.1 Analysis of Copula Marginals

The marginals describe the univariate distribution of CNN features, and are an important component of the copula analysis. ReLU has the effect of zeroing out (deactivating) any negative-valued features such that they have no further impact on the intermediate calculations. In order to fully and adequately describe the 1D marginal distribution post-activation, we must separately model the zero-valued and non-zero features as the combination of these distributions describes the overall marginal density. The percentage of non-zero

Table 1: Percent of nonzero features per layer.

| | ResNet-18 | | | | ResNet-50 | | | | VGG-19 | | | |
|---|---|---|---|---|---|---|---|---|---|---|---|---|
| | INET | CF10 | CF100 | MNST | INET | CF10 | CF100 | MNST | INET | CF10 | CF100 | MNST |
| Layer 0 | 87.7 | 66.4 | 67.2 | 74.8 | 91.0 | 61.3 | 74.8 | 86.0 | 43.2 | 41.9 | 43.2 | 58.0 |
| Layer 1 | 77.0 | 68.1 | 79.6 | 66.7 | 80.1 | 83.9 | 81.6 | 90.8 | 29.6 | 21.1 | 23.8 | 32.4 |
| Layer 2 | 50.3 | 43.7 | 50.9 | 52.9 | 55.8 | 68.0 | 85.2 | 90.8 | 14.8 | 22.0 | 13.2 | 23.3 |
| Layer 3 | 46.1 | 24.0 | 35.6 | 38.4 | 29.9 | 21.7 | 49.6 | 58.5 | 9.9 | 79.1 | 51.3 | 52.1 |
| Layer 4 | 52.2 | 46.8 | 55.0 | 41.3 | 53.9 | 84.9 | 57.9 | 89.6 | 9.0 | 30.6 | 25.8 | 21.5 |

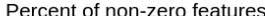

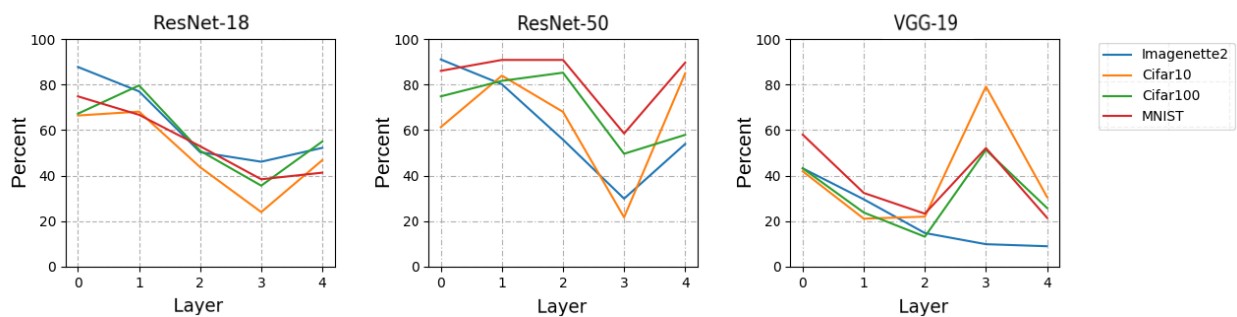

Figure 3: Percent of nonzero features per layer.

features is shown in figure 3 and also in tabular form in table 1. The percent of nonzero features varies for each architecture (ResNet-18, ResNet-50, VGG-19) as well as for each dataset (Imagenette2, CIFAR-10, CIFAR-100, and MNIST). We see however a general trend amongst all models that the percent of nonzero features is quite high in layer 0, and tends to decrease in subsequent layers until layer 3 (ResNet-18, ResNet-50) or layer 2 (VGG-19) before increasing slightly in subsequent layers. One primary exception to this overall shape is shown in VGG-19 over Imagenette2, in which the percent of nonzero features appears to decrease monotonically all the way to layer 4. The ResNet models tend to start with $70 - 90\%$ of nonzero features in layer 0, decreasing to around $50\%$ of nonzero features by layer 4 with the exception of ResNet-50 MNIST, which exhibits $89.6\%$ nonzero features in layer 4. The VGG-19 architectures exhibit greater sparsity than the ResNet-18 architectures with $40 - 60\%$ non-zero features in layer 0 decreasing to $9 - 31\%$ non-zero features in layer 4. Overall these results exhibit an increase in feature sparsity (percent zeros) with network depth, although this trend is somewhat noisy often showing an uptick in nonzero percentage in the deepest layers. The magnitude and scale of these percentages are dataset dependent with VGG-19 showing greater sparsity than ResNet architectures.

Figure 4 shows a histogram of the non-zero portion of the marginal density for ResNet-18 on Imagenette2 for filters 0, 1, and 2. Additional plots of the non-zero marginals for ResNet-18, ResNet-50, and VGG-19 are available in supplemental materials. We see in figure 4 that filters 0 (left), 1 (middle), and 2 (right) show similar shape characteristics that depend on the network depth. Importantly, we observe an interesting phenomenon, where the early layers show a more complicated univariate shape, whereas the later layers appear to more closely resemble an exponential distribution shape. The complicated shape seen in layers 0 and 1 is most likely due to contamination of the input pixel distribution into the marginal feature distribution, as these are early shallow layers in the network. In the early layers, the model is unable to transform the input pixel space very much, so we still observe remnants of this pixel distribution. However, we see in the deeper layers such as layer 3 and 4 a very clear exponential distribution of the non-zero features. We observed a similar exponential distribution consistently across other architectures and datasets with additional similar figures in supplemental materials.

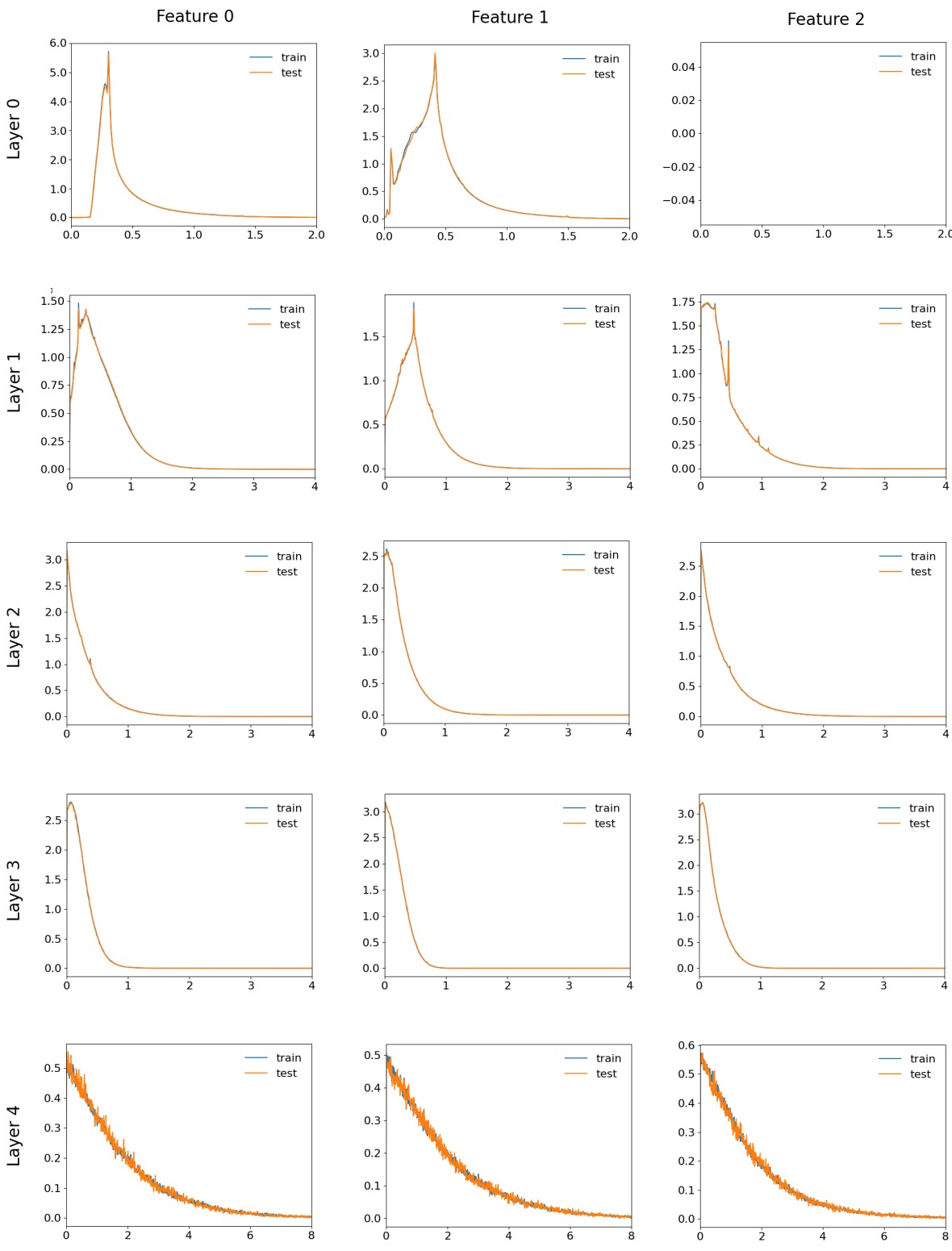

Figure 4: Histogram of marginal density for pre-trained ResNet-18 on Imagenette2 for features 0,1,2, for each of five convolutional layers. In early layers, features show some influence of original pixel distribution. Layer 0 also exhibits a few dead features (e.x. feature 2). Deeper layers appear to exhibit an exponential distribution.

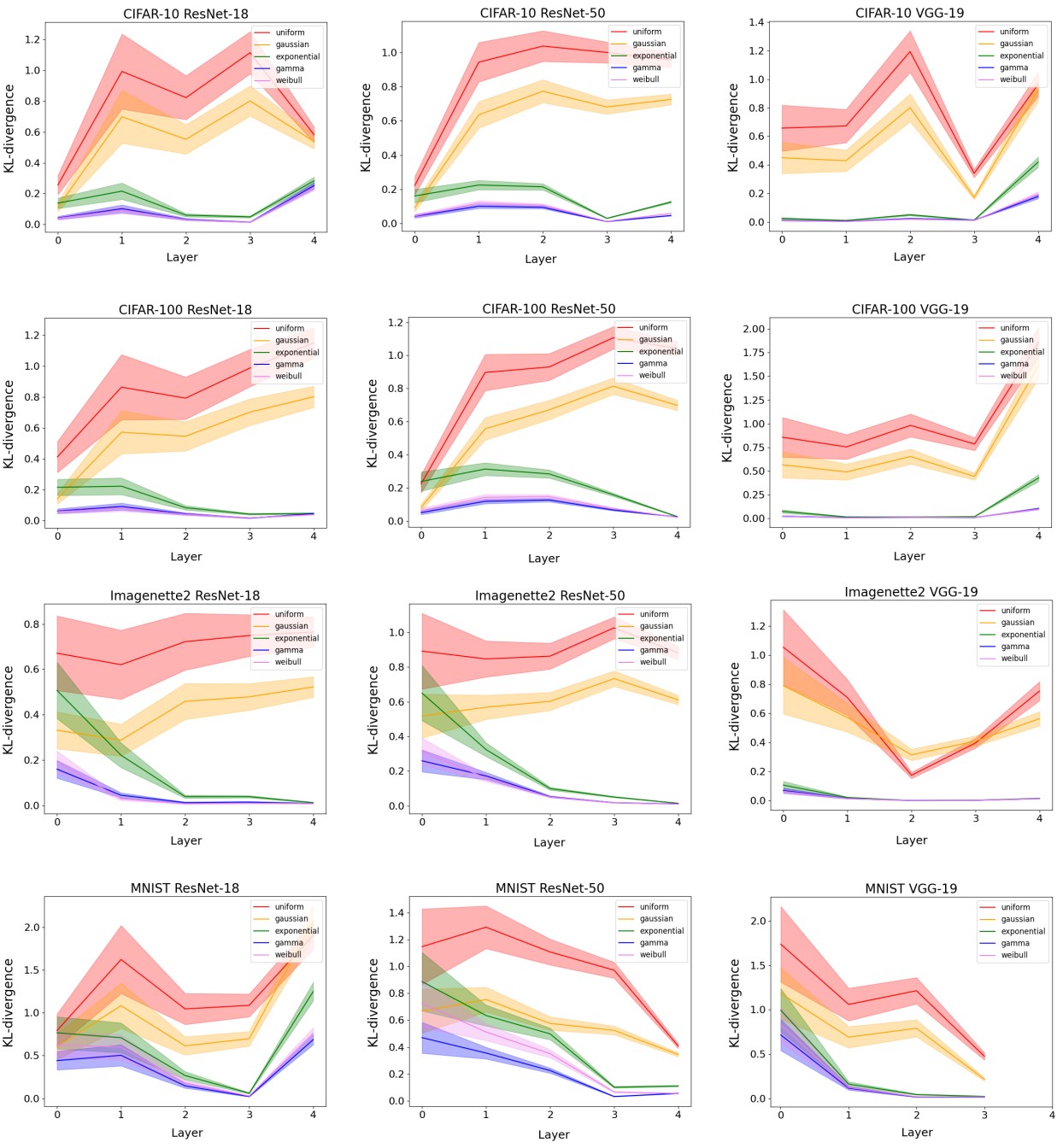

Figure 5: Quantitative goodness-of-fit of five standard distributions to the feature marginals for CIFAR-10, CIFAR-100, Imagenette2, and MNIST, across three models ResNet-18, ResNet-50, and VGG-19. Shaded region shows 95% confidence interval for goodness of fit. We see that exponential, gamma and Weibull are substantially and significantly better fit than uniform and Gaussian for most layers across all models and datasets, with gamma and Weibull showing the best fit but these distributions generalize the exponential distribution which shows increasingly good fit in the deeper layers of the network.

We now quantitatively compare the goodness of fit of five parametric distributions to the marginals of the non-zero features. The distributions that we compare are the uniform distribution, the Gaussian distribution, the gamma distribution, and the Weibull distribution. The optimal parameters of these distributions are determined using the method of stochastic hill climbing, and the goodness of fit of each of the distributions is compared using KL-divergence. Figure 5 shows the plots with 95% confidence intervals of the goodness-of-fit for all five distributions tested over all of the architectures and datasets in our analysis. In order to prevent overfitting, the parametric distribution is fit to the training data features, whereas the KL-divergence measures the goodness of fit to the test distribution. Additional details on the experimental procedure are available in appendix A.

As we see from figure 5, the gamma and Weibull distributions show a significantly better fit to the feature marginals than the uniform and Gaussian distributions. The exponential distribution can be seen as having performance in-between the Gaussian and Weibull distribution. Notably, we observe that the exponential distribution increasingly approximates the observed features as we look at deeper network depth. It is important to note that the gamma and Weibull distributions generalize the exponential distribution, and thus it is not possible for the exponential to achieve a statistically better fit than the gamma and Weibull distributions. Nevertheless, the exponential distribution is highly interpretable and exhibits a very good fit to the non-zero features that significantly outperforms the Gaussian and uniform distributions, while coming close to the gamma and Weibull goodness of fit particularly for the deeper layers.

## 4.2 Analysis of The Optimal Tail Parameter

We performed a numerical analysis of the extreme value behavior of the deep feature marginals in order to determine the extent to which deep CNN features may be considered as long tailed. Our numerical analysis applies the high-threshold technique of Extreme Value Theory (EVT). Using this technique, we defined a threshold equal to the $99^{th}$ percentile of the training sample for any given feature sample. We then numerically fit the Weibull distribution with minimum 1-Wasserstein distance to the high valued sample. The details of this calculation are given in appendix B.

The theoretical analysis of Vladimirova et al. (2019) suggests that calculation of an optimal Weibull tail parameter $\theta$ as a way of determining if the marginals are indeed long-tailed. The Weibull distribution is highly flexible, because it allows for the modeling a range of tail behaviors, where $\theta \leq \frac{1}{2}$ corresponds a to a sub-Gaussian distribution, $\theta \leq 1$ corresponds to a sub-exponential distribution, and $\theta > 1$ is long-tailed.

Figure 6 shows a box plot of the optimal weibull tail parameter $\theta$ layer by layer for a variety of architecture/dataset pairs. For every layer, the box and whiskers define the quantiles of the optimal Weibull $\theta$ tail parameter. We also plot a green line at $\theta = \frac{1}{2}$ and a red line at $\theta = 1$. Firstly, we observe that the deepest layer (layer 4) shows a large dataset complexity dependence. For all models, the optimal $\theta$ for layer 4 is much larger for the complex datasets (CIFAR-100, Imagenette2) than it is for the simpler datasets (CIFAR-10, MNIST). This result occurs without exception in Figure 6, strongly suggesting that the distribution of sub-categories plays a role in the long-tailedness of the deepest conv layer. In literature context, Feldman & Zhang (2020) observed clearly that CIFAR-100 and ImageNet have a more prominent *long-tail* of *rare* instances, than does MNIST. One would expect that CIFAR-100 would also have a longer tail of *rare* instances than CIFAR-10 as is typical of datasets with a larger number of categories (Zhang et al., 2025). Nevertheless, Vladimirova et al. (2019) would suggest that feature marginals become increasingly long-tailed with depth until the very deepest layer, but this is not what we observe in Figure 6. Instead, we observe that the long-tailedness of the deepest layer depends heavily on the complexity of the dataset, with more complex datasets (CIFAR-100, Imagenette2) exhibiting longer optimal $\theta$ values than simpler datasets (CIFAR-10, MNIST).

We henceforth define the *deep intermediate layers* to be layers 1, 2 and 3 for ResNet, and layers 1 and 2 for VGG-19. This definition is logical because, we see that layers 3 and 4 very clearly exhibit this aforementioned dataset complexity dependence for VGG-19, whereas only layer 4 exhibits this dependence for ResNet. We observe that for the median optimal $\theta$ value increases monotonically for the *deep intermediate* for all dataset/model pairs without exception. This strongly suggests, that the phenomenon of increasing long-tailedness with depth is at least partially true. However, the rate of increase is highly data/model dependent,

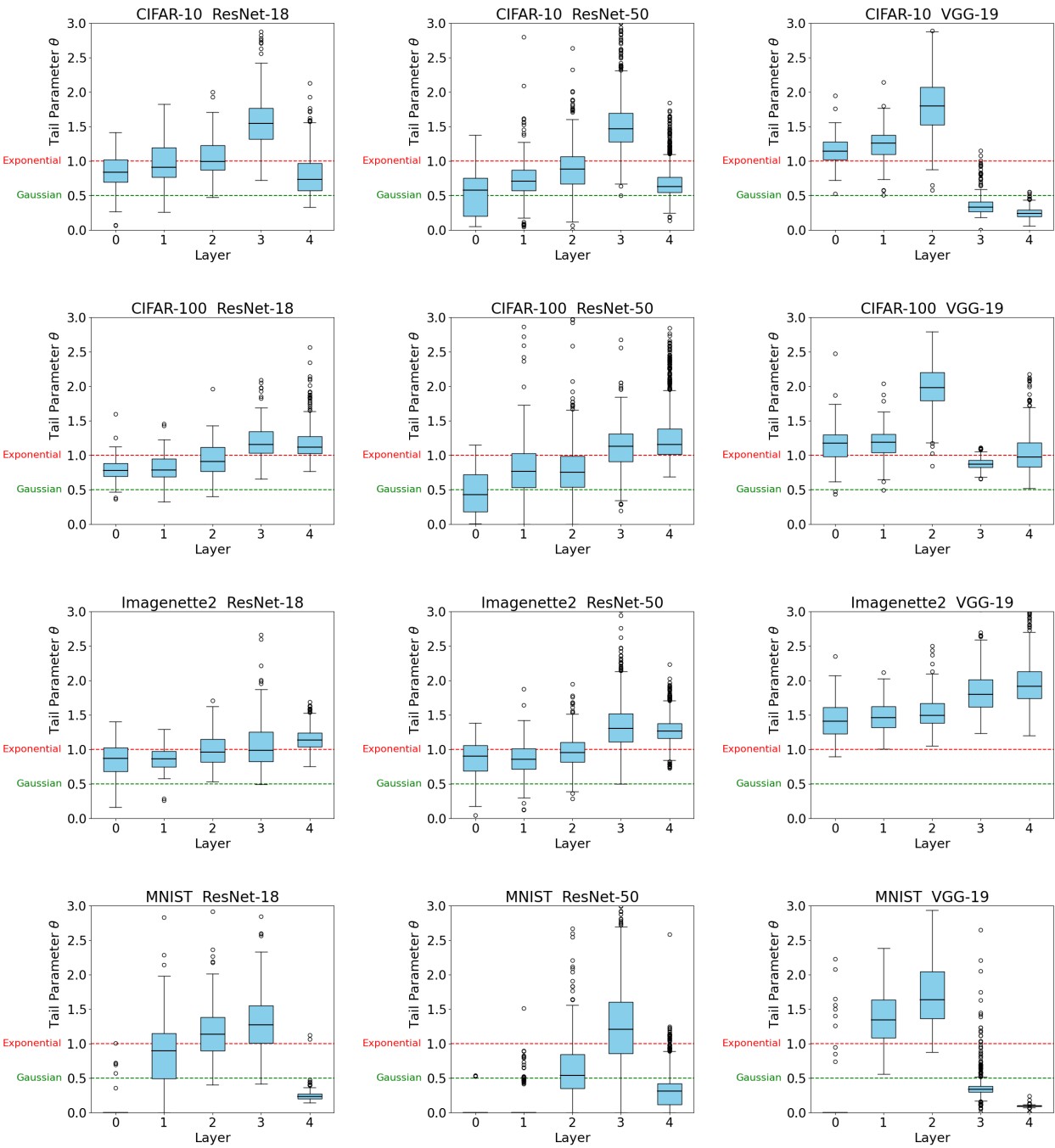

Figure 6: Tail parameter analysis for CIFAR-10, CIFAR-100, Imagenette2, and MNIST, across three models ResNet-18, ResNet-50, and VGG-19.

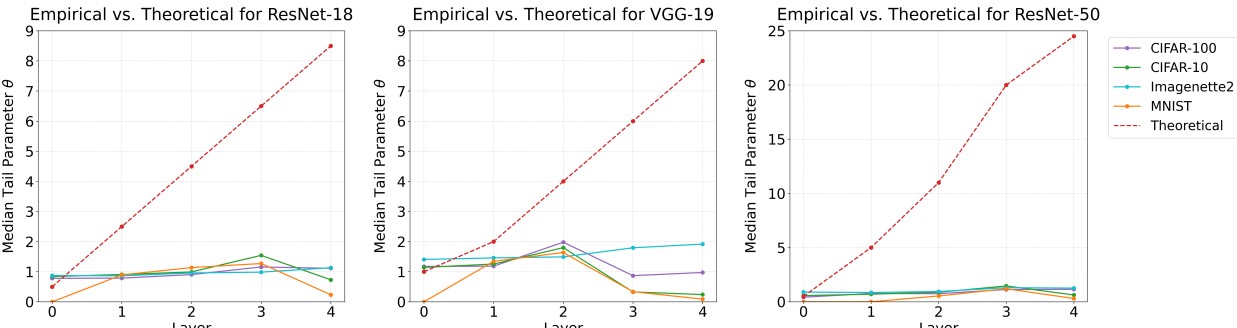

Figure 7: Empirical optimal Weibull $\theta$ tail parameters per layer (solid) versus theoretical estimates (dashed).

and is also substantially slower than predicted by (Vladimirova et al., 2019). The difference between the empirical and theoretical $\theta$ values is shown in a comparison in Figure 7. We see that the differences are indeed very large. Nevertheless, the *deep intermediate* do in fact show a monotonic increase in $\theta$ albeit very slowly. Notice that we define layer as after the major blocks in Figure 2, whereas (Vladimirova et al., 2019) defines layer as after each convolution and/or linear layer which is much more fine grained. The plot of Figure 7 takes this difference in definition into account.

It is observed that for every model and dataset pair, at least one layer is *long-tailed* in that the median optimal $\theta$ parameter is greater than 1. Moreover, all of the datasets and models with the exception of Imagenette2 ResNet-18 exhibit a long tail for the *deepest intermediate* layer (Layer 3 for ResNet and Layer 4 for VGG-19). For Imagenette2 ResNet-18, the *deepest intermediate* layer exhibits a median $\theta$ of 0.985 which is sub-exponential. As such, at least one layer is *long-tailed* in all data/model pairs, and in all but one pair, the *deepest intermediate* layer is long-tailed.

We also observe that VGG-19 has significantly longer tails than comparable ResNet models for the *deep intermediate* layers. Across all datasets, all of the *deep intermediate* layers for VGG-19 exhibit long tails. For ResNet, only the *deepest intermediate* layer is long-tailed, with the exception of MNIST ResNet-18, where Layer 2 is also long-tailed, and Imagenette2 ResNet-18 where all layers are sub-exponential. As such, the *deep intermediate* layers of VGG-19 are noticeably longer-tailed than those of ResNet.

### 4.3 Analysis of Copula Interdependence

Figure 8 shows the copula interdependence term of the probability density for ResNet-18 over Imagenette2 for select pairwise features. Additional plots for other architectures and datasets are available in supplemental materials. Figure 8 also shows the reconstructed copula interdependence using three different methods, the GCF with normalized Legendre polynomials (left), the GCF with real-valued Fourier series (middle) and 2D histograms as a control (right). We see that all three functions appear to show very good agreement with very similar pairwise density plots for all of the feature pairs shown in figure 8.

We observe that the early layers show much more complicated interdependence with greater variation, whereas later layers tend to show a similar structure for most feature pairs that we observed. Similar to our analysis of the marginals, we believe that the variation in the pairwise interdependence seen in the shallow layers (layer 0 and layer 1) is very likely due to the contamination of the input pixel and texture distribution into the shallow features. As these are early layers in the network, the network architecture is unable to fully remove the contribution of the initial pixel distribution. The later layers however show very similar pairwise interdependence. We observe a very interesting phenomenon, in which the deep features in layers 2,3 and 4 (rows 3-5 in figure 8) appear to be uncorrelated for typical values, but show a strong statistical dependence for extremely large valued features. This strong statistical dependence can be observed by the prevalence of very high density in the upper right corner of the interdependence plot (yellow), with much lower density along the top and right edges of the plot (blue). The apparent statistical independence of the terms over typical values (green) is also clearly seen in the pairwise interdependence plots.

In order to evaluate the goodness-of-fit of the copula independence term, we perform a supervised evaluation using cross entropy loss. Our task is to fit the copula interdependence to the training features using our method, and then we evaluate how well this describes the interdependence of the test features. For this experiment, we calculated the goodness of fit for ResNet-18 features for random groups of four features at a time for the Imagenette2, CIFAR-10, CIFAR-100 and MNIST dataset. We performed 30 rounds per experiment, and calculated the copula interdependence using 3 methods, Legendre GCF, Fourier GCF, and Histograms. The interdependence of four randomly selected features within the same layer is calculated, and compared using cross entropy loss.

For ResNet-18 on CIFAR-10, Legendre GCF works better for layers 2 and 3; however, for the latest layer, the Histogram perform best. Table 4 shows that on CIFAR-100, Legendre GCF performs well across all layers except layer 1, where Histogram outperforms. On MNIST (table 5), the Histogram method is the best for all layers except layer 3, where Legendre GCF outperforms. This is because MNIST is essentially a binary dataset with most pixels being either black or white. This binary input pixel distribution contaminates the deep feature distributions, causing a discrete quantization which can be better modeled by the discrete Histogram approach. For the other datasets (Imagenette2, CIFAR-10, CIFAR-100) with more continuous input pixel distributions, we observe in tables 2-4 that the GCF methods significantly outperform the Histogram methods for layers 2,3 and 4, with the two exceptions being layer 3 for Imagenette2 dataset, and layer 4 for the CIFAR-10 dataset. The Histogram approach is particularly sensitive to the CoD, because with $D$ dimensions and $B$ bins per dimension, the average number of samples per bin is $N/B^D$. Due to max-pooling, the deeper layers exhibit fewer samples relative to the shallow layers, and one would expect the GCF methods to outperform the histogram methods particularly in the situations of higher dimensionality as well as smaller sample sizes, which is consistent with our results in tables 2-4.

Table 2: Goodness of fit of copula interdependence on Imagenette2 with ResNet-18. Evaluation criteria is cross entropy loss, 95% confidence intervals are shown in parentheses.

|  | Legendre | Fourier | Histogram |
|---|---|---|---|
| Layer 0 | 1.8329 (1.7882, 1.8775) | 1.8854 (1.8398, 1.9309) | **1.6276 (1.5710, 1.6841)** |
| Layer 1 | 2.6372 (2.6353, 2.6392) | 2.6376 (2.6357, 2.6395) | **2.6238 (2.6213, 2.6264)** |
| Layer 2 | **2.7143 (2.7140, 2.7146)** | 2.7146 (2.7142, 2.7149) | 2.7173 (2.7170, 2.7176) |
| Layer 3 | 2.7375 (2.7374, 2.7376) | **2.7367 (2.7366, 2.7369)** | 2.7405 (2.7403, 2.7407) |
| Layer 4 | **2.7176 (2.7174, 2.7179)** | 2.7197 (2.7195, 2.7199) | 2.7326 (2.7324, 2.7328) |

Table 3: Goodness of fit of copula interdependence term for CIFAR-10 with ResNet-18. Evaluation criteria is cross entropy loss, 95% confidence intervals are shown in parentheses.

|  | Legendre | Fourier | Histogram |
|---|---|---|---|
| Layer 0 | 2.2306 (2.2048, 2.2564) | 2.2359 (2.2103, 2.2616) | **2.1850 (2.1580, 2.2121)** |
| Layer 1 | 2.5844 (2.5816, 2.5873) | 2.5812 (2.5783, 2.5841)) | **2.5578 (2.5519, 2.5637)** |
| Layer 2 | **2.6834 (2.6822, 2.6847)** | 2.6852 (2.6840, 2.6864) | 2.6865 (2.6852, 2.6879) |
| Layer 3 | **2.7442 (2.7438, 2.7446)** | 2.7461 (2.7457, 2.7464) | 2.7493 (2.7490, 2.7496) |
| Layer 4 | 2.1568 (2.1439, 2.1697) | 2.1268 (2.1143, 2.1393) | **2.0399 (2.0282, 2.0516))** |

Table 4: Goodness of fit of copula interdependence term for CIFAR-100 with ResNet-18. Evaluation criteria is cross entropy loss, 95% confidence intervals are shown in parentheses.

|  | Legendre | Fourier | Histogram |
|---|---|---|---|
| Layer 0 | **2.2020 (2.1613, 2.2427)** | 2.2085 (2.1689, 2.2482) | 2.0857 (2.0440, 2.1274) |
| Layer 1 | 2.5588 (2.5554, 2.5622) | 2.5600 (2.5566, 2.5635) | **2.5242 (2.5197, 2.5286)** |
| Layer 2 | **2.6794 (2.6782, 2.6807)** | 2.6806 (2.6794, 2.6818) | 2.6829 (2.6816, 2.6841) |
| Layer 3 | **2.7414 (2.7413, 2.7415)** | 2.7415 (2.7414, 2.7416) | 2.7449 (2.7449, 2.7450) |
| Layer 4 | **2.7200 (2.7197, 2.7203)** | 2.7220 (2.7217, 2.7223) | 2.7303 (2.7300, 2.7306) |

Table 5: Goodness of fit of copula interdependence term for MNIST with ResNet-18. Evaluation criteria is cross entropy loss, 95% confidence intervals are shown in parentheses.

|  | Legendre | Fourier | Histogram |
|---|---|---|---|
| Layer 0 | -3.3750 (-3.9090, -2.8410) | -2.7168 (-3.0019, -2.4318) | **-3.6498 (-4.1857, -3.1139)** |
| Layer 1 | 1.3320 (1.1720, 1.4921) | 1.2019 (1.0059, 1.3979) | **0.5817 (0.2066, 0.9567)** |
| Layer 2 | 2.1437 (2.1159, 2.1716) | 2.1298 (2.0980, 2.1616) | **1.8885 (1.8182, 1.9588)** |
| Layer 3 | **2.6451 (2.6438, 2.6464)** | 2.6551 (2.6539, 2.6563) | 2.6481 (2.6464, 2.6497) |
| Layer 4 | 2.1646 (2.1296, 2.1996) | 2.1572 (2.1188, 2.1956) | **2.0449 (1.9857, 2.1041)** |

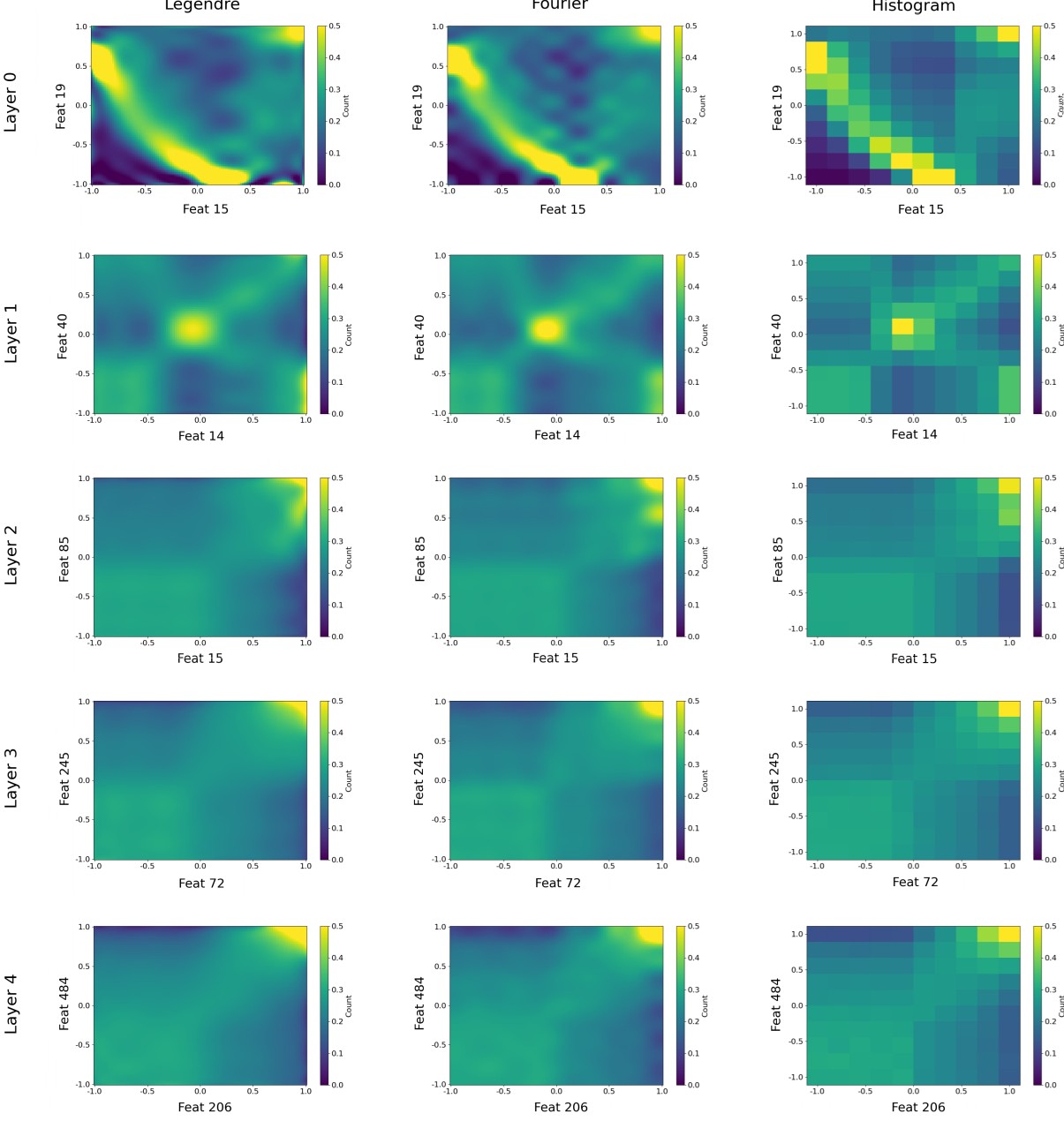

Figure 8: Select copula interdependence for pairwise features for 5 layers of ResNet-18 over Imagenett2. Top to bottom: Layers 0 through 4. Left: Legendre. Middle: Fourier. Right: Histogram.

## 5 Inter-comparison with related methods

We compare our the performance of our novel copula+GCF method against standard Archimedean copula models as well as the standard multivariate Empirical Characteristic Function (ECF) for the task of modeling the bivariate copula density with Cross Entropy loss as an objective criteria. Our supervised task is to estimate the density of the test features given a copula or PDF model from the training features. Additional details for this task are given in appendix D.

Archimedean copulas are defined as any copula $C$ of the following form, where $\Psi$ is a *Generator* function, and $\theta$ is an interdependence hyperparameter.

$$C\left(y_1,\, y_2,\, \ldots,\, y_D\,;\, \theta\right) \quad = \quad \Psi^{-1}\big(\Psi\left(y_1;\theta\right);\, +\, \Psi\left(y_2;\theta\right);\, +\, \ldots\, +\, \Psi\left(y_D;\theta\right)\,;\, \theta\big) \tag{29}$$

Within the Archimedean family, there are many choices for the generator function $\Psi$. We compare against the Gumbel (Gumbel, 1960), Frank Frank (1979), Clayton (Clayton, 1978), Ali Mikhail and Haq (AMH) (Ali et al., 1978), and Joe (Joe, 1994) generator functions $\Phi$, as these are the most popular and well studied generators for practical applications. In the bivariate case, the hyperparameter $\theta$ is derived using a generator-specific formula based on either Kendall's $\tau$ or Spearman's $\rho$ which are calculated from the training data. Additional details for the training of the Archimedean baseline are given in appendix D.

Additionally we compare our copula+GCF method against the standard ECF technique (McCulloch, 1996; Yu, 2004; Nolan, 2013). The ECF is considered to be a powerful method for estimating complex multivariate probability density functions. For a sample of $N$ $D$-dimensional values $X \in \mathbb{R}^{D \times N}$ one can define the frequency parameter $k \in \mathbb{R}^D$. The multivariate ECF $\phi(k) : \mathbb{R}^D \to \mathbb{R}$ is the sample approximation of the multivariate characteristic function, and is defined as the following.

$$\phi(k) \;=\; \frac{1}{N}\, \sum_{n=1}^{N}\, \prod_{d=1}^{D}\, e^{i\, k_d\, X_{d,n}} \tag{30}$$

The PDF is therefore reconstructed as a multivariate inverse Fourier transform of the ECF as follows.

$$f\left(X\right) \;=\; \frac{1}{(2\pi)^d}\, \int_{\mathbb{R}^D}\, \prod_{d=1}^{D}\, e^{i\, k_d\, X_{d,n}}\, \phi(k)\, dk \tag{31}$$

It is standard practice to convolve the ECF basis functions with a smoothing kernel in order to improve the sampling of $\mathbb{R}^D$ in Eq (31). We applied a triangular smoothing kernel with additional details given in appendix D. The smoothing kernel helps somewhat, but numerically sampling of $\mathbb{R}^D$ remains problematic.

We found the ECF baseline to be unstable, and difficult to tune relative to our copula+GCF method. There is no standard procedure for how to sample the numeric integral over $\mathbb{R}^D$ in Eq (31), only rules of thumb that are a starting point for further manual tuning (see appendix D). In order to obtain reasonable results for the ECF baseline, we had to employ $16\times$ additional moments relative to our method, a substantial increase in computation. Moreover we had to separately manually tune the sampling of $\mathbb{R}^D$ for every dataset and model layer in order to obtain reasonable results. Tables 6 - 9 show that our copula+GCF method greatly outperforms the ECF baseline, even if we afford the ECF baseline these unfair advantages. Given a level playing field, the copula+GCF method would almost certainly outperform the ECF baseline by an even wider margin than what we report.

Tables 6 - 9 show the performance of our copula+GCF technique versus the ECF and Archimedean baselines for ResNet-18 over the Imagenette2, CIFAR-10, CIFAR-100 and MNIST datasets. We see that in all cases, our method with Legendre and Fourier moments greatly outperforms the ECF and Archimedean techniques. Moreover we see that the ECF and Archimedean copulas appear to show comparable results for deeper layers, whereas Archimedean outperforms ECF for the shallow layers. However, these results are far below the reported accuracy of the Legendre and Fourier copula+GCF techniques. We performed a t-test (details

in appendix C) which shows that copula+GCF with either Legendre or Fourier significantly improves the cross entropy loss of the test features relative to ECF and Archimedean baselines for all datasets and layers shown in Tables 6 - 9 without exception. Moreover this improvement passes a 99.9% confidence interval in cases for this intercomparison.

Figure 9 shows a comparison of copula interdependence for all of the methods in this comparison. We see that the Legendre and Fourier show very detailed probability density functions, whereas all of the Archimedean copulas show oversmoothed density plots. This oversmoothness is due to the limited parameterization of the Archimedean copulas, which prevent these methods from learning complex inter-dependencies. We also see that the ECF shows some details similar to the Legendre and Fourier, but also vast *dead-zones* (dark) where the ECF underestimates the density. The primary reason for these *dead-zones* is because the ECF does not separate marginal and interdependence terms in the same way that a copula model might do so. The ECF attempts to fit the Fourier transform of feature PDF directly, and in doing so it has difficulty in fitting the univariate marginal terms. The true marginals are non-negative, possibly long-tailed, and exhibit a discontinuous density at zero. Given an infinite number of terms, the characteristic function can represent any probability distribution by means of its Fourier transform. Unfortunately, the ECF cannot be sampled infinitely, and the Fourier transform is known to require an impractically large number of terms to adequately approximate functions with power law asymptotes, which is a problem for long-tailed marginals. Moreover, the discontinuity around zero due to ReLU may require an impractically large number of Fourier terms due to Gibbs phenomenon. Furthermore, as the characteristic function is continuous, a practitioner still has to decide on which frequencies to actually draw the samples, which based on our experience is non-trivial and required substantial and tedious manual tuning.

By comparison, our copula+GCF method uses copula in order to remove the contribution of the marginals, which are not suitable for empirically sampling of the characteristic function. This has a profound impact, because it frees up the GCF to focus solely on the interdependence which does not exhibit any difficult long-tails or discontinuities. Moreover, our GCF method is based on a discreet countable series of orthogonal functions, such as the Legendre polynomials or our proposed discreet real-valued Fourier series. For example, with the Legendre polynomials, it is very clear that one should sample the zeroth order polynomial (constant), then first order (linear), then second order (quadratic), and so on up to a desired order $K$. We set the same hyperparameter $K = 11$ for all experiments without needing to retune this parameter. This is a large practical advantage over the ECF, which mixes the marginal and interdependence terms together and provides little guidance of how best to sample the continuous Fourier series.

Table 6: Intercomparison with related methods for random bivariate subsets on Imagenette2 with ResNet-18. Evaluation criterion is cross entropy loss of copula density. Bold indicates the best method for each layer.

|  | Layer 0 | Layer 1 | Layer 2 | Layer 3 | Layer 4 |
|---|---|---|---|---|---|
| **Legendre** | 1.1883 | 1.3757 | **1.3794** | **1.3823** | **1.3711** |
| **Fourier** | **1.1878** | **1.3748** | 1.3796 | 1.3825 | 1.3716 |
| ECF | 2.1029 | 2.0450 | 2.1529 | 2.0635 | 1.4537 |
| AMH | 1.5329 | 1.4933 | 1.4953 | 1.4969 | 1.5002 |
| Clayton | 1.4469 | 1.4756 | 1.4755 | 1.4806 | 1.4631 |
| Frank | 1.4779 | 1.4818 | 1.4825 | 1.4869 | 1.4762 |
| Gumbel | 1.4749 | 1.4836 | 1.4845 | 1.4888 | 1.4808 |
| Joe | 1.5020 | 1.4867 | 1.4883 | 1.4917 | 1.4890 |

Table 7: Intercomparison with related methods for random bivariate subsets on CIFAR-100 with ResNet-18. Evaluation criterion is cross entropy loss of copula density. Bold indicates the best method for each layer.

|            | Layer 0    | Layer 1    | Layer 2    | Layer 3    | Layer 4    |
|------------|------------|------------|------------|------------|------------|
| **Legendre** | **1.2740** | **1.3635** | **1.3757** | **1.3782** | **1.3618** |
| **Fourier**  | 1.2799     | 1.3640     | 1.3759     | 1.3784     | 1.3660     |
| ECF        | 1.4692     | 1.8947     | 1.6868     | 1.4477     | 1.5810     |
| AMH        | 1.5043     | 1.5003     | 1.4993     | 1.4992     | 1.5024     |
| Clayton    | 1.455      | 1.4793     | 1.4746     | 1.4756     | 1.4763     |
| Frank      | 1.4688     | 1.4865     | 1.4833     | 1.4843     | 1.4845     |
| Gumbel     | 1.4740     | 1.4893     | 1.4863     | 1.4871     | 1.4876     |
| Joe        | 1.4861     | 1.4937     | 1.4914     | 1.4918     | 1.493      |

Table 8: Intercomparison with related methods for random bivariate subsets on CIFAR-10 with ResNet-18. Evaluation criterion is cross entropy loss of copula density. Bold indicates the best method for each layer.

|            | Layer 0    | Layer 1    | Layer 2    | Layer 3    | Layer 4    |
|------------|------------|------------|------------|------------|------------|
| **Legendre** | **1.2843** | **1.3604** | **1.3761** | **1.3781** | 1.1741     |
| **Fourier**  | 1.2888     | 1.3610     | 1.3763     | 1.3783     | **1.1699** |
| ECF        | 1.5974     | 1.9448     | 1.5633     | 1.4555     | 1.4363     |
| AMH        | 1.5061     | 1.5005     | 1.5011     | 1.5        | 1.5078     |
| Clayton    | 1.4615     | 1.4711     | 1.48       | 1.4774     | 1.4351     |
| Frank      | 1.4741     | 1.4802     | 1.488      | 1.4857     | 1.4527     |
| Gumbel     | 1.478      | 1.4841     | 1.4903     | 1.4884     | 1.4593     |
| Joe        | 1.4884     | 1.4906     | 1.4947     | 1.493      | 1.4779     |

Table 9: Intercomparison with related methods for random bivariate subsets on MNIST with ResNet-18. Evaluation criterion is cross entropy loss of copula density. Bold indicates the best method for each layer.

|            | Layer 0    | Layer 1    | Layer 2    | Layer 3    | Layer 4    |
|------------|------------|------------|------------|------------|------------|
| **Legendre** | **1.0807** | 0.725      | 1.2268     | **1.3211** | 0.6217     |
| **Fourier**  | 1.1273     | **0.6626** | **1.2232** | 1.3247     | **0.5865** |
| ECF        | 1.399      | 1.3791     | 1.3815     | 1.3809     | 1.2408     |
| AMH        | 1.5068     | 1.5119     | 1.5011     | 1.499      | 1.5275     |
| Clayton    | 1.4641     | 1.4449     | 1.4489     | 1.4583     | 1.3863     |
| Frank      | 1.4756     | 1.4549     | 1.4598     | 1.4687     | 1.4513     |
| Gumbel     | 1.4785     | 1.4583     | 1.4658     | 1.4738     | 1.4018     |
| Joe        | 1.4871     | 1.4725     | 1.4784     | 1.483      | 1.4854     |

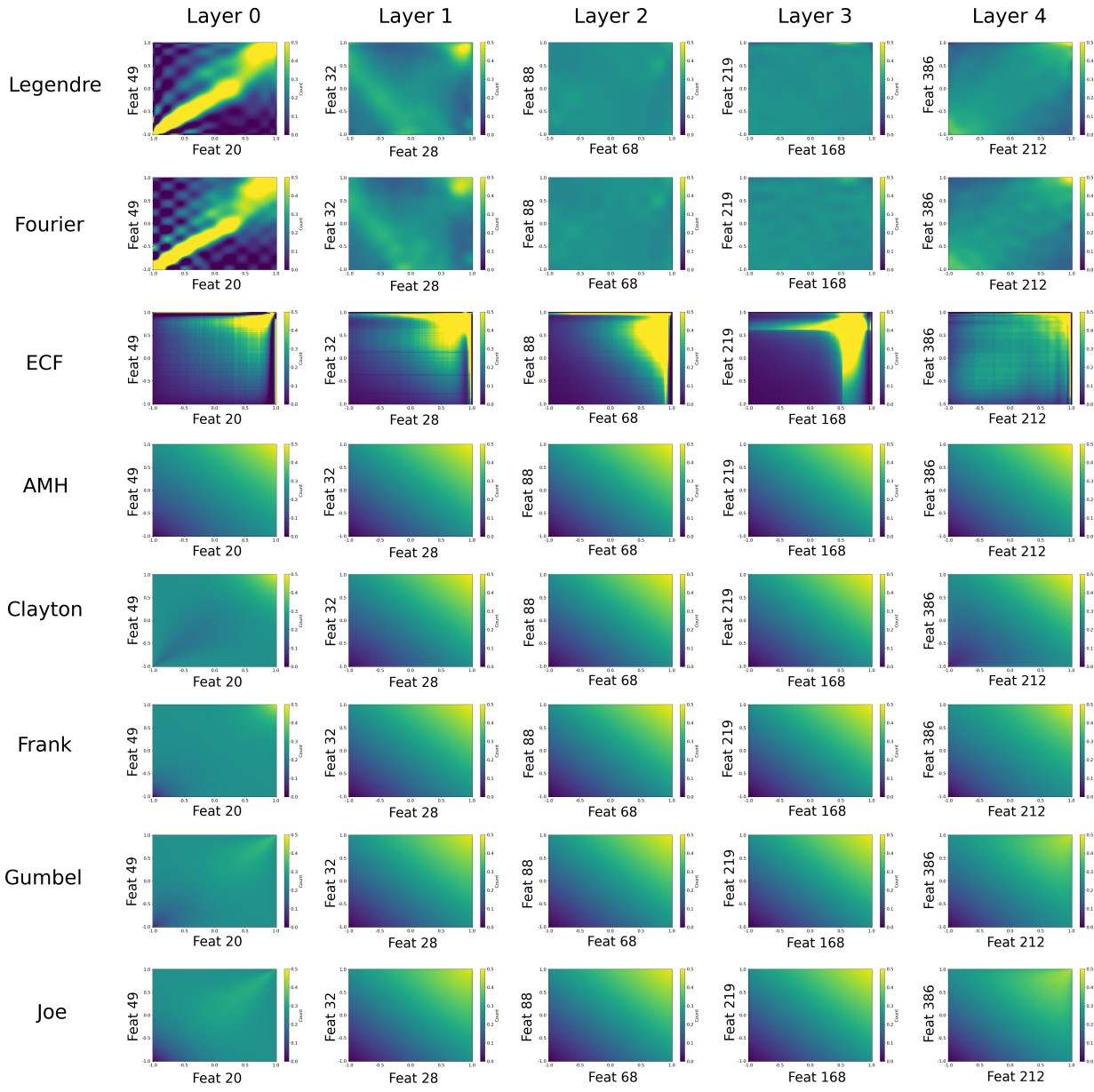

Figure 9: Select copula interdependence for pairwise features for 5 layers of ResNet-18 over Imagenett2. Top to bottom: Legendre, Fourier, ECF, AMH, Clayton, Frank, Gumbel, and Joe. Left to right: Layers 0 through 4.

## 6    Conclusion

We present an empirical analysis of the density distribution of deep CNN features through direct estimation of the GCF using a novel non-parametric approach that combines copula analysis with the MOM. We demonstrate that our approach is able to model the marginal and interdependence terms of feature density after each major Conv+ReLU block of ResNet-18, ResNet-50, and VGG-19. Moreover, as a non-parametric technique, we do not introduce overly-restrictive assumptions as to the shape of the marginal or interdependence terms. We report empirical findings on the observed marginal distributions and copula density interdependence terms as a function of network depth. In our analysis of marginals, we observe that features after major Conv+ReLU blocks exhibit both zero and non-zero features. Furthermore, we demonstrate through hypothesis testing that the non-zero features for the deeper layers of the network more closely fit the Weibull or gamma distribution versus the Gaussian or uniform distribution. Weibull and gamma distributions generalize the exponential distribution which also significantly outperforms the Gaussian and uniform distributions in the deeper layers. Our analysis of marginals also shows a prevalence of *long-tailed* deep features, with every model and dataset pair exhibiting at least one layer with *long-tailed* marginals. However, the length of the tail grows only slowly with depth, and many layers exhibit *sub-exponential* marginal tails Moreover, the length of the tail for the deepest layers shows strong dependence with dataset complexity, as more complex datasets (CIFAR-100, Imagenette2) exhibit longer tails for this layer than simpler datasets (CIFAR-10, MNIST). Our analysis of the copula interdependence shows that pairs of features in the deeper layers of the network also exhibit a form of statistical dependence, for which these features are highly independent throughout their typical value ranges, yet become strongly dependent (either correlated or anti-correlated) for extremely large feature values. We observe that the Legendre and Fourier GCF techniques are able to better model the copula interdependence density relative to the histogram technique for groups of four features with ResNet-18 over Imagenette2, CIFAR-10 and CIFAR-100 for the deeper layers of the network. Our approach is the first purely empirical technique to model the multivariate probability density distribution of deep CNN features. Moreover, our copula+GCF approach greatly outperforms the related Archimedean copula and ECF techniques in terms of its ability to represent deep CNN feature density. We believe that empirical analysis of the feature density distribution will lead to a better understanding of CNN feature representations. Empirical feature density has the potential to lead to a new branch of generative methods that model the full joint distribution of the native CNN feature space, thereby enabling off-the-shelf CNN architectures to attain all of the benefits of generative techniques.

## 7    Discussion

One of our key findings is that the Weibull and Gamma distributions, which both generalize the exponential distribution offer a surprisingly good fit for the non-zero features of the marginals for the deeper layers of the network. This result has many implications for applications of feature density techniques, and also leads to several new questions regarding the reason *why* an exponential and/or long-tailed distribution fits these deeper non-zero marginals so well. One possible explanation is from the computer vision perspective. In the deeper layers, features correspond to increasingly expressive semantic concepts (Allen-Zhu & Li, 2023). If the features represent presence of a discriminative target of interest, then most of the scene does not usually contain the target. For example, given a face in a cluttered scene, the vast majority of the scene does not contain the face (and would exhibit a small-value for face detection). A small portion of the scene however does present a face, and a face detector would present a strong finding. It is possible that in the deeper layers, features correspond to detectors of semantic concepts, in which case each large valued feature would correspond to a strong concept detection.

We also observe that all of the neural networks under consideration exhibit at least one deep feature layer with *long-tailed* marginals. This is an important finding, because it is now understood that the frequency of object subcategories follows a long tail, and that even *class balanced* datasets exhibit a *long-tail* of rare instances (Zhang et al., 2025; Feldman, 2020; Feldman & Zhang, 2020). Nevertheless, we also observe that not all deep feature layers are long-tailed, in fact many of the more shallow intermediate layers exhibit sub-exponential tails. This observation disagrees with theoretical analysis of Bayesian networks suggesting that the *long-tail* of marginals grows very rapidly with depth (Vladimirova et al., 2019). The theoretical result

depends on the assumption of Gaussian weights and imagery which is unrealistic as it does not describe the behavior of a well trained network on natural imagery. Based on our results we conjecture that *training* of a CNN using natural imagery may play a role in *suppressing* the length of the marginal tails. Investigation of this conjecture would be an interesting problem for future research.

We also observed a very peculiar phenomenon where pairs of deep CNN features at adequate network depth show very little dependence for *typical values*, whereas they exhibit very strong dependence (either correlation or anticorrelation) for extremely strong detections. We conjecture that this distribution that we see is highly related to our *long-tailed view* hypothesis, in that the *typical values* of the features correspond to a lack of detection of the *target view*, whereas the extremely large values correspond to a strong detection of the *target view*. As such, it may be the case that the *typical values* are uncorrelated because they do not represent discriminative foreground features, but instead represent background variability especially as most images have a large number of background pixels relative to foreground pixels. The rarity of discriminative foreground features would therefore be a reason for CNN models to produce very large detections in order to overcome the more prevalent background signal.

If one believes in this *long-tailed view* hypothesis, then the strong statistical dependence of extremely large feature values would imply that in fact deep CNN features are more correlated than they appear in the presence of actual foreground target detection signal, even if they are uncorrelated over the background. This is very possible and we plan to investigate this hypothesis further as part of future work.

Moreover, the uncorrelated nature of *typical value* features, yet strong correlation of *extreme value* features, would suggest that future work should revisit the assumption that the *typical set* is in fact the most descriptive set of features for a given image, as modeling of feature density has overwhelmingly emphasized evaluation of the density distribution of *typical valued* features. We believe that modeling of the distribution of *extreme valued* features may also be highly important. Moreover, the removal of outlier features under the assumption of an MVG has the potential to accidentally eliminate strong computer vision signals of foreground target detection which would be an unwanted side effect.

Our approach toward modeling the interdependence of features by measuring the GCF with the MOM has a notable advantage that the entire training set is used to measure each sample moment, leading to high certainty in estimation. This is a major inherent advantage in modeling the density distribution of high-dimensional feature spaces, versus other techniques that run into low-sample sizes in high dimensions due to the CoD. Nevertheless, this technique also has limitations. Notably, the number of moments increases exponentially with the number of features under consideration at once. Not all moments are inherently important, and future work would involve the extension of this method to find a set of sparse moments, in order to reduce the computational burden of modeling the copula interdependence for a large number of features within one subset group. An additional limitation of this methodology is the error-of-approximation that comes from reconstructing the density using sample moments rather than population moments. We infer that this error-of-approximation is reasonably small by looking at the cross entropy goodness of fit between the train and test distributions. However, in future work, we would like to statistically model the uncertainty of the moment estimates, as this would allow us to quantify the uncertainty of the empirical probability density estimate.

Our inter-comparison with Archimedean copulas, and multivariate ECF techniques reveals that our proposed copula+GCF method has large advantages over both of these prior methods. Notably Archimedean copulas are quite inflexible and prone to underfitting the true complexities of the interdependence. This is because the Archimedean copulas are based on only a single rank-shape statistic. By comparison, our approach uses many rank-shape statistics including Spearman's $\rho$ (which corresponds to the linear Legendre term), but also an entire series of orthogonal moments that could be considered as rank-shape statistics. This is a major advantage in modeling complex inter-dependencies between variables such as deep CNN features.

We were also surprised to see that our proposed copula+GCF technique greatly outperforms the multivaraite ECF in terms of its ability to model the interdependence of deep CNN features. Copula analysis uses the probability integral transform to model the univariate marginals, which is a very expressive representation for univariate distributions. The probability integral transform can easily tolerate the long-tailed and discontinuous (around zero) distributions that we observed in the deep feature marginals. By comparison the ECF

requires and impractical number of *sin* and *cos* frequencies to adequately fit the marginals, and this failure with the marginal terms further defeats the ability of the ECF to adequately learn the interdependence terms. Our GCF approach also presents a discreet ordered sampling of orthogonal functions, rather than a continuous set of Fourier frequencies as in the ECF, thereby alleviating concerns about how best to sample the frequencies. This straightforward sampling strategy further eliminates the need for extensive manual tuning. As such, our proposed copula+GCF technique may have practical advantages over classical methods in representing the complex interdependence of confounding variables, and may find new applications both within and outside of the field of deep learning.

**Acknowledgments**

Anonymous Acknowledgment

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

## A   Appendix. Detailed Experimental Design of Marginal Goodness of Fit

**Optimal Distribution Parameters**

To model the univariate marginal distributions of non-zero CNN feature activations, we evaluate five parametric distributions including uniform, Gaussian, gamma, Weibull, and exponential. Each distribution is fit individually to the empirical histogram of non-zero feature values using the training set using the method of stochastic hill climbing. For a given parametric distribution $q_\theta(x)$ with parameters $\theta$, we minimize the KL-divergence between the empirical distribution $p(x)$ estimated via histogram and the parametric model.

$$\text{KL}(p\|q_\theta) = \sum_{j=1}^{B} p(j) \log\left(\frac{p(j)}{q_\theta(j)}\right) \tag{32}$$

Where $j$ is the histogram bin index of, $p(j)$ is the empirical probability density of bin $j$, and $q_\theta(j)$ is the value of the parametric PDF evaluated at the bin center. We performed 1000 iterations of *stochastic hill climbing* with a falloff parameter of 0.97. At each iteration, a stochastic candidate model is proposed with a random offset at most the step size from the previous best model, and the step size is multiplicatively reduced using the falloff parameter. The new candidate model accepted if the solution improves the objective criteria.

**Evaluation of Parametric Fit**

After fitting, the parametric models are evaluated using the KL-divergence between the parametric fit and the test histogram. As such, the task presented is to determine how well each of the parametric models fit to the training histogram can approximate the empirical distribution of test samples as measured by a histogram. The shaded regions present 95% confidence intervals of the layer-by-layer KL-divergence as estimated using Student's t-test. A breakdown of the steps involved with this testing procedures are as follows.

1. Compute the KL-divergence for the non-zeros samples of each filter $d$ within the target layer of $D$ filters, We denote this KL-divergence as $\text{KL}_d$. This value measures how well the trained parametric model explains the test histogram of filter $d$.

2. Compute the sample mean of KL-divergence values across all non-zero features in the layer.

$$\overline{\text{KL}} = \frac{1}{D} \sum_{d=1}^{D} \text{KL}_d \tag{33}$$

3. Estimate the standard error of the mean (SE), which quantifies the uncertainty in the estimated average KL-divergence where $s$ is the sample standard deviation of KL-divergences,

$$s = \frac{\sigma}{\sqrt{D}}, \quad \text{where } \sigma = \sqrt{\frac{1}{D-1} \sum_{d=1}^{D} (\text{KL}_d - \overline{\text{KL}})^2} \tag{34}$$

4. Construct a 95% confidence interval (CI) around the mean. Since $N \geq 64$, we approximate the $t$-distribution with the standard normal distribution.

$$\text{CI} = \overline{\text{KL}} \pm z_{0.975} \cdot s, \quad \text{with } z_{0.975} \approx 1.96 \tag{35}$$

These intervals are visualized as shaded bands around the mean KL-divergence values in Figure 5. They indicate the uncertainty in the average KL-divergence for each fitted distribution and enable statistical comparison across distributions and layers. Overlapping intervals suggest no significant difference, while non-overlapping intervals indicate a statistically significant difference in goodness-of-fit.

## B  Appendix. Detailed Experimental Design for Tail Parameters

This appendix provides a detailed description of the experimental design for the analysis of the optimal Weibull tail parameter $\theta$. The PDF and CDF of the Weibull distribution is defined as the following.

$$f(x; \theta, k) = \begin{cases} \theta k \, (\theta x)^{k-1} \, e^{-(\theta x)^k} & x \geq 0 \\ 0 & x < 0 \end{cases} \tag{36}$$

$$F(x; \theta, k) = \begin{cases} 1 - e^{-(\theta x)^k} & x \geq 0 \\ 0 & x < 0 \end{cases} \tag{37}$$

Where $\theta$ is the tail parameter, and $k$ is the shape parameter. The Weibull distribution is often equivalently parameterized using $\lambda = 1/\theta$ known as a scale parameter. But the $\theta$ parameterization is common in Extreme Value Theory (EVT) and matches the parameterization as used in Vladimirova et al. (2019).

Given a univariate sample $X$, in order to numerically calculate the optimal tail parameter, we further define a high value threshold $u$. For tail analysis, we only consider samples that exceed this threshold. We calculate the high value threshold $u$ as the $99^{th}$ percentile of the sample $X$. The $99^{th}$ percentile is a common choice for EVT analysis, and was determined to be reasonable after a visual inspection of the dataset.

$$\text{Define} \quad u \quad \text{s.t.} \quad Pr[X \leq u] = 0.99 \tag{38}$$

Define $\ddot{x} \subseteq X$ as the subset of sample $X$, such that all of the samples exceed high threshold $u$.

$$\ddot{x} = \{ \ddot{x}_i : \ddot{x}_i \in X \text{ and } \ddot{x}_i > u \} \tag{39}$$

Define $\ddot{y}$ as the empirical CDF of $\ddot{x}$, i.e. the corresponding percentiles.

$$\ddot{y}_i = Pr[X \leq \ddot{x}_i] \tag{40}$$

The 1-Wasserstein distance is a well behaved distance metric for comparing partial distributions, and is a good candidate for tail analysis. In the univaraite continuous case, the 1-Wasserstein distance simplifies to minimizing the absolute distance between the CDF of the partial sample and the CDF of the parametric distribution. The focus on aligning CDFs is what makes this loss function appealing for EVT, as it ensures that the high valued percentiles are well aligned between the empirical and parametric distributions within the tail. The overall objective function for fitting the Weibull distribution including tail parameter $\theta$ and shape parameter $k$ is as follows. Note this sum is only calculated over the high valued sample $\ddot{x} \subseteq x$.

$$\underset{\theta, \, k}{argmin} \; \sum_{i=1}^{n} \left|\left| \, \ddot{y}_i - F(\, \ddot{x}_i \, ; \theta, k) \, \right|\right|_1 \tag{41}$$

This optimization was performed in the $C$ programming language by using the method of stochastic hill climbing. We performed 1000 iterations with a falloff of 0.97. In order to speed up the calculation, we sorted sample $x$ and binned the data into 10000 quantiles of equal density. Thus, the largest 100 of these 10000 quantiles constitute the $99^{th}$ percentile threby implementing the high value threshold $u$. These high value quantiles were used for numerical optimization of the Weibull parameters to minimize the 1-Wasserstein loss.

## C   Appendix. Detailed Experimental Design of Copula Interdependence

This appendix provides a detailed description of the methodology used in analysis of copula interdependence.

### Separate Processing for Training and Testing

We have extracted training features from the training data and testing features from the testing data. First, we obtain the empirical marginal and interdependence terms strictly from the training data. Once we obtain these terms, we evaluate our model of copula interdependence by determining how well it fits the probability density of the withheld test features by using the criteria of cross entropy loss.

**Probability Integral Transform**

For any given univariate training and testing set $x_{train}, x_{test}$, we define the samples as the following.

$$
\begin{aligned}
x_{train,i} &\quad \text{as the i}^{th} \text{ training sample for } i \in [0, N_{train} - 1] \\
x_{test,j} &\quad \text{as the j}^{th} \text{ testing sample for } j \in [0, N_{test} - 1]
\end{aligned}
\tag{42}
$$

We apply the empirical Probability Integral Transform (PIT) to resample the post-ReLU features into a uniform marginal distribution. A challenge arises when applying the probability integral transform to post-ReLU features, which often contain a large number of zero-valued entries. This leads to a degeneracy in the CDF due to many tied values, causing ambiguity in the ranking and flattening the transformation. To address this ambiguity, we introduce an infinitesimal random jitter to all zero-valued features during both training and testing. Formally, for any zero-valued feature $x$, the following equations define $\ddot{x}_{\text{train}}$ and $\ddot{x}_{\text{test}}$.

$$
\ddot{x}_{\text{train}} = x_{\text{train}} + \varepsilon, \quad \ddot{x}_{\text{test}} = x_{\text{test}} + \varepsilon
\tag{43}
$$

To prevent data leakage this resampling is performed using the training set. Define the rank $r_{train,i}$ as the rank of the $i^{th}$ training feature as follows.

$$
r_{train,i} \quad \text{the number of training samples } x \in \ddot{x}_{train} \text{ s.t. } x \le \ddot{x}_{train,i}
\tag{44}
$$

As such, we approximate the rank of the testing samples by using the largest rank of any training sample with equal or lesser value to the test sample. This can be computed using binary search, given a sorted training set.

$$
r_{test,j} = sup\{r_{train,i} : x_{train,i} < x_{test,j}\} \frac{N_{test} - 1}{N_{train} - 1}
\tag{45}
$$

In our implementation, the empirical PIT is calculated by sorting all of the training features in the range $[0, n-1]$ in order to obtain a set of $n$ ordered ranks. Typically, the probability integral transform converts a marginal distribution into a uniform distribution over the interval $(0, 1)$. However, in our approach, we carry out the analysis using a rescaled version of the probability integral transform that maps the feature values to the interval $(-1, 1)$. This rescaling is motivated by the fact that many standard orthogonal functions are defined on this interval, allowing us to represent the copula density in a richer and more flexible way without parametric assumptions. The modified probability integral transform is defined in the following equation.

$$
F_i(x) = 2 \cdot \Pr[X_i \le x] - 1
\tag{46}
$$

or equivalently, following shows using the rank $r$ to get the CDF of each training feature.

$$
F_i(x_{train,i}) = 2 \cdot \left( \frac{r_{train,i}}{N_{train} - 1} \right) - 1 \qquad F_i(x_{test,i}) = 2 \cdot \left( \frac{r_{test,i}}{N_{test} - 1} \right) - 1
\tag{47}
$$

**Copula Density and Its Evaluation**

The empirical moments $\hat{\mu}$ and copula density $\hat{c}$ are calculated used the algorithm as detailed in the pseudocode and formal definition. The moments are calculated using the procedure CALCULATEINTERDEPENDENCE over $D$ features, which are a random subset of the filters within a given layer for analysis. This procedure is implemented as a $C$ program that takes as input the entire training sample for the specified $D$ features, and outputs a set of $K^D$ empirical moments known as $\hat{\mu}$. As such the empirical moments are computed entirely from the training set. This set of moments further fully defines a model of the copula interdependence $\hat{c} : \mathbb{R}^D \to \mathbb{R}$ which is the dot product of the moments and the set of orthogonal functions.

$$
\hat{c}(\vec{y}) \quad = \quad \hat{\mu} \cdot \Phi(y) \quad = \sum_{T \in \mathbb{Z}_K^D} \hat{\mu}_T \, \Phi(\vec{y})
\tag{48}
$$

Now that our copula interdependence model is $\hat{c}$ is estimated from the training data, our task is to evaluate how well it models the probability density of the features from the test set. Cross entropy loss is used for the evaluation criteria.

For a given set of test features $X_{test}$, the transformed test features $Y_{test}$ are calculated using the probability integral transform. Then, we use our model $\hat{c}$ to determine the predicted probability of the test features. This predicted probability is compared against the true probability of $1/N_{test}$ because empirically each test feature is equally likelihood. Therefore, the overall cross entropy evaluation is calculated using the following summation over the test set.

$$\text{Cross-Entropy} \; = \; -\frac{1}{N_{\text{test}}} \sum_{y \in Y_{test}} \log\left(\hat{c}(y)\right) \tag{49}$$

**Confidence Intervals**

The entire training and testing process is repeated 30 times for each model, dataset, and layer using a different subset of $D$ features. For this analysis we used $D = 4$. By repeating this process 30 times, we straightforwardly calculate 95% confidence intervals using Student's t-test for the reported cross entropy loss statistics.

# D    Appendix. Detailed Experimental Design of Inter-comparison

We discuss our inter-comparison with the Archimedean copulas and the ECF. Our evaluation criteria remains as Cross-Entropy of the copula density as described in Appendix C. The primary difference is that for the inter-comparison, we focus on the bivariate analysis and discarded any feature pairs for which Spearman's $\rho$ and Kendall's $\tau$ are negative. This is because the Archimedean copulas are designed primary for bivariate analysis, and many of the Generators have restricted ranges for Spearman's $\rho$ or Kendall's $\tau$.

**Archimedean copulas**

Archimedean copulas are the most common copulas in applied statistics. These copulas make use of a Generator function $\Psi(y; \theta)$ that is invertible. As there is only one hyperparameter parameter $\theta$, typically Archimedean copula's are used in the bivariate case, where the overall copula is as follows.

$$C(y_1, y_2, \theta) \quad = \quad \Psi^{-1}\big(\; \Psi(y_1; \theta) \; + \; \Psi(y_2; \theta) \; ; \; \theta \;\big) \tag{50}$$

The Gumbel (Gumbel, 1960), Frank Frank (1979), Clayton (Clayton, 1978), Ali Mikhail and Haq (AMH) (Ali et al., 1978), and Joe (Joe, 1994) generator functions are the most popular choices. These generators are defined as follows.

$$
\begin{aligned}
\text{Gumbel:} \quad & \Psi(y; \theta) = t^{\theta}, & \theta \geq 1 \\[2mm]
\text{Clayton:} \quad & \Psi(y; \theta) = \frac{1}{\theta}(y^{-\theta} - 1), & \theta > 0 \\[2mm]
\text{AMH:} \quad & \Psi(y; \theta) = \frac{1 - \theta}{y - \theta}, & -1 \leq \theta < 1 \\[2mm]
\text{Joe:} \quad & \Psi(y; \theta) = -\frac{1}{\log(1 - (1 - \theta)(1 - e^{-y}))}, & \theta \geq 1 \\[2mm]
\text{Frank:} \quad & \Psi(y; \theta) = -\frac{1}{\theta} \log\left(\frac{e^{-\theta y} - 1}{e^{-\theta} - 1}\right), & \theta \neq 0
\end{aligned}
\tag{51}
$$

The hyperparameter $\theta$ can be determined using a formula based on either Spearman's $\rho$ or Kendall's $\tau$ of the bivariate series. For Gumbel, Clayton and AMH, this formula is of closed form. For Joe and Frank only

the inverse formula is closed form. Fortunately, these inverses are strictly monotonic, so one can numerically compute $\theta$ in terms of $\tau$ to high precision using binary search.

$$
\begin{aligned}
\text{Gumbel:} \quad & \theta = \frac{1}{1-\tau} & 0 \le \tau < 1 \\[2mm]
\text{Clayton:} \quad & \theta = \frac{2\tau}{1-\tau} & 0 \le \tau < 1 \\[2mm]
\text{AMH:} \quad & \theta = \frac{3\rho}{3+\rho} & -0.2711 < \rho < 0.4784 \\[2mm]
\text{Joe:} \quad & \tau = 1 - 4 \sum_{k-1}^{\infty} \frac{1}{k\left(\theta k + 2\right)\left(\theta(k-1)+2\right)} & 0 \le \tau < 1 \\[2mm]
\text{Frank:} \quad & \tau = 1 - \frac{4}{\theta}\left(1 - \int_0^{\theta} \frac{t}{e^t - 1} dt\right) & -1 < \tau < 1
\end{aligned}
\tag{52}
$$

### D.1 Empirical Characteristic Function (ECF)

The PDF is calculated from the multivariate empirical Empirical Characteristic Function using a triangular smoothing kernel of bandwidth $b$ as follows.

$$
f(X) \approx \frac{1}{(2\pi)^d} \int_{\mathbb{R}^D} \prod_{d=1}^{D} \ddot{G}_{k_d,b}(X_{d,n})\, \phi(k)\, dk
\tag{53}
$$

Define $G_t(x) : \mathbb{R} \to \mathbb{R}$ as the univariate complex exponential series.

$$
G_t(x) = e^{itx}
\tag{54}
$$

Define $T_b(z) : \mathbb{R} \to \mathbb{R}$ as the equal area triangular kernel over the interval $(-b, b)$.

$$
T_b(z) = \begin{cases} \frac{1}{b^2} z + \frac{1}{b} & -b < z \le 0 \\ -\frac{1}{b^2} z + \frac{1}{b} & 0 \le z < b \\ 0 & \text{otherwise} \end{cases}
\tag{55}
$$

Define $\ddot{G}_{t,b}(x) : \mathbb{R} \to \mathbb{R}$ as the triangular kernel convolved with the complex exponential to produce a smoothed basis function.

$$
\ddot{G}_{t,b}(x) = \int_{-b}^{b} T_b(z)\, G_t(x+z)\, dz
\tag{56}
$$

One can substitute (54), and perform basic algebra to to obtain the following, where the integral does not depend on x. We can consolidate this integral by defining ancillary variable $J$

$$
\ddot{G}_{t,b}(x) = e^{itx} \int_{-b}^{b} T_b(z)\, e^{itz}\, dz = e^{itx} J
\tag{57}
$$

As the triangular kernel is piecewise, we must split our finite integral $J$ along the piecewise boundaries.

$$
J = \frac{1}{b^2} \int_{-b}^{0} z\, e^{itz}\, dz - \frac{1}{b^2} \int_{0}^{b} z\, e^{itz}\, dz + \frac{1}{b} \int_{-b}^{b} e^{itz}\, dz
\tag{58}
$$

The following identities can be easily derived using integration by parts.

$$\int e^{itz}dz = -i\frac{1}{t}\,e^{itz}\,+\,C \qquad \int z\,e^{itz}dz = \frac{1}{t^2}\,e^{itz}\,-\,i\frac{z}{t}\,e^{itz}\,+\,C \tag{59}$$

Therefore $J$ can be evaluated as follows.

$$J \;=\; \frac{1}{b^2}\left[\frac{1}{t^2}\,e^{itz}-i\frac{z}{t}\,e^{itz}\right]_{-b}^{0} - \frac{1}{b^2}\left[\frac{1}{t^2}\,e^{itz}-i\frac{z}{t}\,e^{itz}\right]_{0}^{b} + \frac{1}{b}\left[-i\frac{1}{t}\,e^{itz}\right]_{-b}^{b} \tag{60}$$

Collecting like terms and simplifying.

$$J \;=\; \frac{1}{b^2t^2}-\frac{1}{b^2t^2}\,e^{-itb}-\frac{1}{b^2t^2}\,e^{itb} \tag{61}$$

Substituting back into (56) and simplifying we obtain

$$\ddot{G}_{t,b}(x) \;=\; \frac{2\,(1-cos(tb))}{b^2t^2}e^{itx} \tag{62}$$

**Experimental considerations**

Our task is largely similar to the evaluation described in appendix C, but with a few minor adaptations to facilitate inter-comparison with Archimedean copulas and ECF functions.

Firstly, for each dataset and model layer, we randomly sampled 30 pairs of features. Unlike the evaluation task of appendix C, for this experiment we first compute Spearman's $\rho$ and Kendall's $\tau$, and ensure that they are within the valid range of $[0, 1)$, because most Archimedean copulas expect rank correlations within this range. Most features pairs are within this range, but if we selected a pair that is not, then we randomly selected a different pair until 30 pairs were picked. The AMH copula has the additional consideration that $\tau$ cannot exceed 0.4784, if $\tau$ exceeded this amount it was clipped at 0.4784 only for AMH, but the exact larger $\tau$ was used for all other Archimedean copulas.

Secondly, we performed our inter-comparison using cross entropy loss within the copula density space $\hat{c}$ as described in appendix C. For Archimedean copulas this is straightforward because they natively project probability densities into this space. For ECF we had to convert density from the PDF space $f(x)$ into the copula density space $\hat{c}(y)$ to facilitate inter-comparison with the copula methods. The reprojection is quite simple with the general formula as follows.

$$\hat{c}(y) = f(x)\prod_{d=1}^{D}\frac{dx_d}{dy_d} \tag{63}$$

The rate of change $dx_d/dy_d$ can is easily obtained using the probability integral transform. For any sample $x_d$, we obtain the quantile $y_d$ using the transform $y_d = F_d(x_d)$. We can invert this monotonic function using binary search to obtain the function $x_d = F^{-1}(y_d)$.

We estimate the the rate of change $dx_d/dy_d$ using the method of finite differences. It is most numerically stable to approximate $dy_d$ as a small fixed delta in quantile space. We then obtain $dx_d$ as the associated change in $x$ as the finite difference using the midpoint rule.

$$dy_d = \Delta \qquad dx_d = \frac{F^{-1}\left(y_d + \Delta\right) - F^{-1}\left(y_d - \Delta\right)}{2} \tag{64}$$

**ECF Sampling considerations**

The ECF requires sampling of an integral over the continuous space $\mathbb{R}^D$. This sampling was not trivial, and required extensive manual tuning. As a general sampling strategy, we sampled a uniform grid of frequencies in each dimension. We used a common rule of thumb that the lowest frequency in each dimension was $\overline{x_d} - h\sigma_d$ and the highest frequency was $\overline{x_d} + h\sigma_d$. We used 44 frequencies in each dimension. As the evaluation task is bivariate, we employed 1936 frequencies total for this experiment. This is $16\times$ the number of moments required by the copula+GCF technique, and thus an additional computational burden. We set the bandwidth $b$ for the triangular smoothing kernel to be equal to the cell-size of the uniform sampling grid. This sampling strategy is a reasonable framework in line with related literature.

The manual tuning was necessary to find adequate values of the parameter $h$. I.e. how many standard deviations from the mean would be reasonable to sample frequencies for the ECF? If $h$ is too small, then the ECF may be overly smooth, whereas if $h$ is too large the ECF may exhibit excessive aliasing. We found that reasonable value of $h$ for one dataset and model layer did not correspond to a reasonable value of $h$ for another. It is known that the characteristic function for a reasonable probability distribution should somewhat visually resemble the *sync* function. We used frequency space plots as a visual aid to double check that any given $h$ was producing a reasonable ECF. We consider the difficulty in manual tuning, and the need for a large number of samples to be disadvantages of the ECF baseline that our copula+GCF approach is able to largely overcome.

# E   Appendix. Orthogonal Function, and Multivariate Counter Pseudocode

The algorithmic design makes use of orthogonal functions $\phi$ and a multivariate counter T. The pseudocode for these utilities is as follows.

---
**Algorithm 4** Evaluation of orthogonal functions for $\vec{y}$

---
1: **procedure** EVALORTHOGONAL($\vec{y}$)
2:     **for** $d \leftarrow 1 \ldots D$ **do**
3:         **for** $t \leftarrow 0 \ldots K$ **do**
4:             $\phi_{d,t} \leftarrow \begin{cases} \dfrac{P_t(\vec{y}_d)}{||P_t||_2}, & \textit{Legendre} \\[2em] cos\left(t\dfrac{\pi}{2}(\vec{y}_d)\right) & \textit{Fourier} \end{cases}$
5:         **end for**
6:     **end for**
7:     **return** $\phi$
8: **end procedure**

---

---

**Algorithm 5** Multivariate Counter Utility Functions

---

1: **procedure** INITIALIZE( )
2:     **for** $i \leftarrow 1 \ldots D$ **do**
3:         $T_i \leftarrow 0$
4:     **end for**
5:     **return** $T$
6: **end procedure**
7:
8: **procedure** INCREMENT$(T)$
9:     $i \leftarrow 1$
10:     $T_i \leftarrow T_i + 1$
11:     **while** $i \leq D$ **and** $T_i \geq K$ **do**
12:         $T_i \leftarrow 0$
13:         $i \leftarrow i + 1$
14:     **end while**
15:     **return** $T$ **if** $i \leq D$ **else** overflow
16: **end procedure**
17:
18: **procedure** ORDERK(T)
19:     sum $= 0$
20:     **for** $i \leftarrow 1 \ldots D$ **do**
21:         $sum = sum + T_i$
22:     **end for**
23:     **return** true **if** $sum < K$ **else** false
24: **end procedure**

---

