# OpenReview forum: "Interpretable Estimation of CNN Deep Feature Density using Copula and the Generalized Characteristic Function"
_TMLR — Rejected by TMLR_

### Review · Reviewer_iNQw · 2025-03-16

**Summary Of Contributions:**

This paper presents an empirical method for measuring the probability density function (PDF) of deep convolutional neural network (CNN) features. The method integrates copula analysis with the Method of Orthogonal Moments (MOM) to estimate the Generalized Characteristic Function (GCF), providing a non-parametric and assumption-free measurement of feature density.

**Audience:**

Yes

**Broader Impact Concerns:**

There are no direct ethical concerns

**Claims And Evidence:**

Yes

**Requested Changes:**

1. Clarify Novelty vs. Prior Work

    Address how the exponential view hypothesis differs from existing research on long-tailed distributions ([1,3,4]).
    Explain the advantage of MOM over PCA-based approaches used in filter pruning ([2]).
    Compare findings to orthogonal projection-based continual learning methods ([5]).

2. Provide a More Direct Comparison to PCA-Based Methods

    The paper should compare MOM-based feature analysis to PCA approaches to demonstrate its benefits or unique insights.
    Include experimental results or citations that highlight the differences between MOM and PCA for filter pruning.

3. Stronger Connection to Practical Applications

    How can exponential distribution of deep features help with model pruning, anomaly detection, or continual learning?
    Would this approach help in designing CNN architectures differently (e.g., improving ReLU activations or feature sparsity)?
    Discussion on potential trade-offs between feature density estimation and network efficiency.

**Strengths And Weaknesses:**

Strengths:

*  Empirical Approach: The use of copula analysis and MOM provides a fresh, assumption-free method for deep feature analysis.
* Challenge to Gaussian Assumption: The paper convincingly demonstrates that deep CNN features do not follow a Gaussian distribution, which has implications for out-of-distribution (OOD) detection and generative modeling.
*  Experimental Validation: The paper tests various architectures (ResNet18, ResNet50, VGG19) and datasets (MNIST, CIFAR10, CIFAR100, Imagenette2), making results more generalizable.

Weaknesses:

* Core Hypothesis is Not Entirely Novel:     The Exponential View Hypothesis aligns with existing research on long-tailed distributions (see Feldman [1,3,4]).
    Prior work has analyzed CNN feature distributions in structured pruning via PCA (see Garg [2]) and continual learning using orthogonal transformations (see Saha [5]).

* Legendre Polynomial and Orthogonality is Not New: The use of orthogonal activation spaces for dimensionality reduction and pruning has been explored using PCA and other moment-based methods (see [2,5]).
    The paper should clarify how MOM differs from PCA-based pruning approaches.

* Limited Discussion on Practical Applications: While the paper presents potential applications, it lacks discussion on practical applications in model pruning, OOD detection, or adversarial robustness.
    It would be valuable to connect their findings to practical implications for CNN architecture design.

* More Comparison Needed to Prior Work: The authors should discuss how their density estimation approach compares to prior statistical models.

[1] Feldman, Vitaly. "Does learning require memorization? a short tale about a long tail." In Proceedings of the 52nd Annual ACM SIGACT Symposium on Theory of Computing, pp. 954-959. 2020.

[2] Garg, Isha, Priyadarshini Panda, and Kaushik Roy. "A low effort approach to structured CNN design using PCA." IEEE Access 8 (2019): 1347-1360.

[3] Feldman, Vitaly, and Chiyuan Zhang. "What neural networks memorize and why: Discovering the long tail via influence estimation." Advances in Neural Information Processing Systems 33 (2020): 2881-2891.

[4] Zhang, Chongsheng, George Almpanidis, Gaojuan Fan, Binquan Deng, Yanbo Zhang, Ji Liu, Aouaidjia Kamel, Paolo Soda, and João Gama. "A systematic review on long-tailed learning." IEEE Transactions on Neural Networks and Learning Systems (2025).

[5] Saha, Gobinda, Isha Garg, and Kaushik Roy. "Gradient Projection Memory for Continual Learning." In International Conference on Learning Representations 2020.

---

> ### Author Response · Authors · 2025-03-31
> **Response to iNQw**
>
> **Strengths
> 1.**  *Empirical Approach: The use of copula analysis and MOM provides a fresh, assumption-free method for deep feature analysis.*
> **2.**  *Challenge to Gaussian Assumption: The paper convincingly demonstrates that deep CNN features do not follow a Gaussian distribution, which has implications for out-of-distribution (OOD) detection and generative modeling.*
> **3.**  *Experimental Validation: The paper tests various architectures (ResNet18, ResNet50, VGG19) and datasets (MNIST, CIFAR10, CIFAR100, Imagenette2), making results more generalizable.
> Weaknesses:*
>
> We appreciate your sentiment and share your enthusiasm that assumption free methods for deep feature analysis will allow us to better understand the behavior of the distribution of deep CNN features.  We are hopeful that the empirical discoveries using this method will lead to new and better theories toward modeling deep feature distributions, in addition to having a practical impact on the development of well justified probability density techniques for downstream tasks.
>
>
> **Weakness 1.**  *Core Hypothesis is Not Entirely Novel: The Exponential View Hypothesis aligns with existing research on long-tailed distributions (see Feldman [1,3,4]). Prior work has analyzed CNN feature distributions in structured pruning via PCA (see Garg [2]) and continual learning using orthogonal transformations (see Saha [5]).*
>
>
> We truly appreciate this insight, and it leads to a wonderful question!  How can we improve our analysis of marginals in order to better contribute toward the literature on long-tailed feature distributions?  The 'exponential view' hypothesis, although it appearing well aligned, is imprecise, especially as exponential distributions are not quite long-tailed.
>
> We have nearly completed a major new experiment to enhance our analysis of marginals to determine if the empirical distribution is in fact truly 'Exponential', or is it actually 'Long-tailed' or even 'Sub-exponential'?  Moreover, how does the length of the empirical optimal Weibull tail parameter $\theta$ change with layer depth.
>
> Reviewer FLH4 had identified a very intriguing paper of Vladimirova (2019).  The paper proves theoretically that Bayesian CNNs with Gaussian training data exhibit increasingly long tails with layer depth, exhibiting optimal Weibull tail parameter $1/2 i$ where $i$ is the layer.  We want to know if this theoretical phenominon depends on oversimplistic assumptions, or if there is empirical evidence that this phenominon fore realistic model architectures and datasets.  We will revise our 'exponential view' hypothesis to better reflect this more detailed analysis.
>
>
> **Weakness 2.**  *Legendre Polynomial and Orthogonality is Not New: The use of orthogonal activation spaces for dimensionality reduction and pruning has been explored using PCA and other moment-based methods (see [2,5]). The paper should clarify how MOM differs from PCA-based pruning approaches.*
>
> Thank you, and we will be certain to expand our literature section to discuss the related moment and linear pruning strateges.  We have reviewed the papers that you suggest as well as several others in this area, and we are confident that we can clearly articulate the differences between our proposed method, and many of the related techniques that are used in feature representation.  We will be including this expanded literature review in our updated version of the paper within 1 week.
>
>
>
> **Weakness 3.**  *Limited Discussion on Practical Applications: While the paper presents potential applications, it lacks discussion on practical applications in model pruning, OOD detection, or adversarial robustness. It would be valuable to connect their findings to practical implications for CNN architecture design.*
>
> We greatly appreciate this feedback, and we will include a much expanded version of the related literature, and we will emphasize density based OOD detection, along with other methods that rely on density modules.  Notably, several existing methods could potentially use our technique as an enhanced drop-in replacement for the feature density module.  We think that it would be good to go into some detail regarding where this technique could be used immediately.
>
> **Weakness 4.**  *More Comparison Needed to Prior Work: The authors should discuss how their density estimation approach compares to prior statistical models.*
>
> We take this constructive feedback very seriously.  Toward this goal, we are working on a second major revision (est 3 weeks) in which expand our analysis of the interdependence term in comparison versus other existing methods.  We talk about the proposed major revisions in greater detail on the overall response to all reviewers.  Our experiment will compare the performance of our novel copula versus standard copula methods as well as advanced feature density techniques including characteristic functions.

---

### Review · Reviewer_oT6W · 2025-03-17

**Summary Of Contributions:**

The authors model the signals in a deep CNNs using a copula-based model with exponential marginals. They claim that such a model is helpful in understanding deep representations, downstream applications such as anomlay detection, and is also interpretable. They compare their exponential model fits against other marginal parametric models.

**Audience:**

No

**Broader Impact Concerns:**

No broader impact concerns.

**Claims And Evidence:**

No

**Requested Changes:**

- Please clarify how this work relates to a usual notion of "interpretable" in machine learning.
- Please clarify whether the model makes parametric assumptions or not.
- Did you try modelling the signals supported on $\mathbb{R}$ instead of just those on $(0, \infty)$? Would a Gaussian here do well compared with an exponential?
- Please clarify the role of maximum entropy.
- Please show a conceptual or practical downstream application of this work.

**Strengths And Weaknesses:**

- Interpretable is mentioned in the title and several times in the paper. However, the authors do not give a definition of "interpretable", and fail to convince me that any of the empirical analyses performed relate to usual notions of "interpretable" machine learning. Figure 6 shows couplings between what appears to be a subset of arbitrary layers, and while these individual couplings can be interpreted, the network as a whole remains just as much of a black box as before I saw the figures. The text does not aid in interpretation of the results.
- The authors mention that their approach is non-parametric and does not introduce overly restrictive modelling assumptions. Yet, one of their main findings is that the exponential distribution fits the marginals well, and they only know this by comparing various parametric family goodness of fits. This seems like a parametric model with restrictive modelling assumptions.
- The exponential distribution is a good fit to the nonnegative signals in the network, and as expected it outperforms Gaussian when the signals are rectified. However, most of the useful uses of the Gaussian assumption in related literature relates to pre-activation signals (which may be positive or negative). Did you try modelling these signals as well?
- In the discussion, the authors offer that the reason the exponential model might fit the data well is because it arises as a maximum entropy distribution with nonnegative support and specified mean parameter (they fail to mention the second constraint of a specified mean parameter). However, there are many other maximum entropy distributions in exponential families. There are many other maximum entropy distributions with nonnegative support and additional moment constraints (e.g. Truncated normal, Rayleigh, ...) see https://en.wikipedia.org/wiki/Maximum_entropy_probability_distribution#Positive_and_specified_mean:_the_exponential_distribution . Furthermore, they do not convincingly argue that maximum entropy is in any way related to training CNNs.
- I do not see any downstream applications demonstrated in this paper (e.g. anomaly detection). I wonder whether it is possible to use any of these findings for any practical or conceptual takeaway?

Minor:
- I find the terminology "measuring a PDF" (similar to abstract, and throughout) a bit strange. Measuring a function is a bit of an unusual setting, and I suspect the authors are trying to convey something different here. Maybe "estimating" or "approximating"?
- The text in Figure 1 is too small to read

---

> ### Author Response · Authors · 2025-03-31
> **Response to oT6w**
>
> **Weakness 1.**  *Interpretable is mentioned in the title and several times in the paper. However, the authors do not give a definition of "interpretable", and fail to convince me that any of the empirical analyses performed relate to usual notions of "interpretable" machine learning.*
>
> Thank you very much for this important feedback.
>
> You raise a very important point, that our method does not provide insight into the semantic meaning of the individual features, as is a more traditional ML problem.  We will clarify the writing.
>
> By 'interpretable', we are referring to the notion that it is possible to visually gain insight into the shape of the deep feature PDF.  It is very difficult for a human to understand anything high-dimensional, much less confounding deep feature statistics.  Copula analysis, as a statistical tool can aid in the interpretation of high dimensional probability densities by separating the marginal and inter-dependence terms to aid in visualization and density estimation.
>
> **Weakness 2.**  *The authors mention that their approach is non-parametric and does not introduce overly restrictive modelling assumptions. Yet, one of their main findings is that the exponential distribution fits the marginals well, and they only know this by comparing various parametric family goodness of fits. This seems like a parametric model with restrictive modelling assumptions.*
>
> It is true that we 'verified' the finding parametrically.  But we did not 'discover' the finding in this way.  The 'discovery' was made through visualization.  The non-parametric copula separates the marginal and inter-dependence terms for visualization and density estimation.
>
> The finding of an exponential fit is of practical importance because it is a very simple approximation that works well for the deep layers.  Some downstream tasks may only need a simple parametric model that is fast and easy to integrate.  In this case an exponential fit may be adequate.  But other tasks may require a much more accurate model of the high-dimensional feature distribution, that is both predictive and visualizable.  In this case, the full copula technique would be more appropriate.
>
> We will be adding major revisions, in order to expand this analysis of the marginals, to look at the Weibull tail parameter.  This new analysis will help us improve our finding on the marginal distribution will also putting it in the context of prior work on long tailed feature distributions.
>
>
> **Weakness 3.**  *The exponential distribution is a good fit to the nonnegative signals in the network, and as expected it outperforms Gaussian when the signals are rectified. However, most of the useful uses of the Gaussian assumption in related literature relates to pre-activation signals (which may be positive or negative). Did you try modelling these signals as well?*
>
> We did not perform experiments, although we did visualize these features and you are right that the pre-activation distributions appear closer to a Gaussian.  Although interestingly, for deep layers the peak of the distribution is always negative.  The pre-activation plots also may exhibit positive skew.  We focused on the features after activation, because the negative features are immediately truncated to zero.  We will include some additional plots within the paper, and add a discussion regarding the pre-relu features.
>
>
> **Weakness 4.**  *In the discussion, the authors offer that the reason the exponential model might fit the data well is because it arises as a maximum entropy distribution with nonnegative support and specified mean parameter (they fail to mention the second constraint of a specified mean parameter). However, there are many other maximum entropy distributions in exponential families.*
>
> We take this concern very seriously, and given the existent literature on 'long-tailed' features, it makes much sense for us to address this point comprehensively through major revisions.
>
> 1.  Greatly enhance related literature discussion to address prior work on long-tailed feature distributions
>
> 2.  Expand analysis of marginals in order to estimate the optimal Weibull tail parameter, and compare this analysis with the theoretical results of Vladimirova under different assumptions.
>
> 3.  Replace the 'exponential view' hypothesis with an updated and more precise hypothesis rooted in the literature and our new analysis.
>
>
> **Weakness 5.**  *I do not see any downstream applications demonstrated in this paper (e.g. anomaly detection). I wonder whether it is possible to use any of these findings for any practical or conceptual takeaway?*
>
> Of the downstream applications mentioned, we believe that Out-of-distribution detection would be the low-hanging fruit that would be easiest to make an impact.  We will greatly enhance our related literature section to include a discussion of the density based OOD methods, including discussion of how these methods could be modified with our technique.

---

> > ### Comment · Reviewer_oT6W · 2025-03-31
> >
> > Thanks for your response. Your suggested revisions sound like a good way forward. Two follow-ups:
> > - Weakness 1: It might be nice to relate your idea of interpretability, "the notion that it is possible to visually gain insight into the shape of the deep feature PDF", to any existing notion of interpretability in the literature (I have not seen this idea of interpretability before, btw, so I cannot suggest any references). Alternatively, try and find some middle ground between your idea of interpretability and more established notions of interpretability (e.g. https://www.nature.com/articles/s42256-019-0048-x)
> > - Weakness 4: I am not seeing the connection between your response (which looks like a good idea, mostly focused on iNQw's comments) and my original concern about maximum entropy distributions. My concern was simply about the relevance of maximum entropy distributions to the work that was done in your paper. Could you clarify this point?

---

### Review · Reviewer_FLh4 · 2025-03-17

**Summary Of Contributions:**

The paper aims to estimate the distribution over activations in CNN architectures and uses an empirical approach based on estimated copulas to do so. For this, the authors define copulas using the generalised characteristic function with unit-integral normalised Legendre polynomials. The authors then inspected the distribution of the marginals by considering the fit of different marginal distributions as well as the interaction terms provided by the copula. The authors find an intriguing property that interaction terms in later layers of CNNs are strongly dependent on the activation values, showing dependencies only for extreme values (in the activation).

**Audience:**

No

**Broader Impact Concerns:**

There are no ethical implications related to this work. Hence no Broader Impact Statement is required.

**Claims And Evidence:**

Yes

**Requested Changes:**

_General Remarks:_

- Missing punctuation in equations. For example, Eq (5) should end with punctuation as it is part of the previous sentence.
- Names of data sets and models generally do not use correct capitalization. For example, mnist should be MNIST, resnet should be ResNet. I recommend checking the original publications for correct capitalization.
- A detailed related work section is missing. For example, I was missing a discussion of related work regarding (i) the distribution of activations in NNs and (ii) the application of copulas in ML. Additional references are listed at the end of this section.

_Detailed Remarks:_

- [p.3] What is the spatial intuition limitation (citation?), and how is it related to the problem? Similarly, the CoD needs justification in this context.
- [p.4] Missing citation for MOM (e.g., [C1])
[p.4] What does "very closely" mean? I recommend either dropping this or making it rigorous.
- [p.5] How is Eq (10) computed? It is unclear how this is achieved. Is this done using numerical approximation? If so, what is the approximation error?
- [p.6] How do Eq (11) and (12) relate? Can you provide more details?
- [p.6] Typo in "[...] we new define [...]"
- [p.6] A formal justification for Eq (15) is needed.
- [p.7] Missing references for ResNet and VGG models.
- [p.7-9] How exactly the copula is defined is unclear. What exactly is the formal definition used in this work? Also, how is it estimated? Only on page 10 is an informal definition given. I recommend providing a clear, formal definition of the model used in this work before the experiments section.
- [p.8] The illustrations in Figure 2 take a whole page but do not convey much information. Consider moving those to the supplement or at least substantially reducing their size. Later illustrations are also somewhat too large for the content they convey. It would be better to use the space to explain the experimental setup, model, etc., in detail.
- [p.9] The experiment in Tab. 1 and Figure 3 seem somewhat arbitrary. Why is the fraction of non-zero activations relevant? Considering quantiles or other more informative measures would be much more informative.
- [p.10] What was the rationale for choosing the specific marginals? Note that it is known that the distribution of activations for Bayesian NN (assuming Gaussian distributed weights) follows a sub-Weibull distribution. [C2]
- [p.10] How the KL divergence was calculated is unknown, as is the "true" distribution. Further details are needed here.
- [p.10] How is the 1-std interval obtained?
- [p.14] What statistical test was used for Tables on page 14? What are the details of the test?
- [p.15] Typo "discreet"

_Additional References:_

List of additional references. Note that there are likely more relevant works that should be discussed.

- [C1] L. L. Yudell. (1975).
Mathematical functions and approximations. Academics Press Inc.
- [C2] Vladimirova, M., Verbeek, J., Mesejo, P., & Arbel, J. (2019). Understanding priors in Bayesian neural networks at the unit level. In International Conference on Machine Learning. PMLR.
- [C3] Ansari, A. F., Scarlett, J., & Soh, H. (2020). A characteristic function approach to deep implicit generative modeling. In Proceedings of the IEEE/CVF conference on computer vision and pattern recognition.
- [C4] Yu, Z., Trapp, M., & Kersting, K. (2023). Characteristic circuits. Advances in Neural Information Processing Systems.
- [C5] Kolouri, S., Martin, C. E., & Hoffmann, H. (2017). Explaining distributed neural activations via unsupervised learning. In Proceedings of the IEEE conference on computer vision and pattern recognition workshops.
- [C6] Zhang, Q., Wu, Y. N., & Zhu, S. C. (2018). Interpretable convolutional neural networks. In Proceedings of the IEEE conference on computer vision and pattern recognition.

**Strengths And Weaknesses:**

**Strengths:**

1. The paper tackles an interesting problem in explainable AI, which could provide insights into model architecture design and failure modes of predictive models.
2. The authors provide a flexible copula design that could be interesting outside of the work.

**Weaknesses:**

1. The paper does not thoroughly evaluate the non-standard copula model used in this work.
2. The evaluation of the distribution over activations in CNNs is only empirical and thus limited in its applicability.
3. The paper lacks a thorough discussion of related work and a precise formal definition of the method itself.
4. The results are very limited and, even though potentially interesting to some, might be insufficient for publication at TMLR.

---

> ### Author Response · Authors · 2025-03-31
> **Response to FLh4**
>
> **Strength 1.**   *The paper tackles an interesting problem in explainable AI, which could provide insights into model architecture design and failure modes of predictive models.*
>
> Thank you for your consideration of the importance of this problem, we want to understand empirically the distribution of deep CNN features in order to provide greater human oversight for models and ensure trustworthy behaviors.
>
>
> **Strength 2.**  *The authors provide a flexible copula design that could be interesting outside of the work.*
>
> Thank you, and we are also be very interested to apply this new copula design to other problems that require high-dimensional probability distributions.
>
>
> **Weakness 1.**  *The paper does not thoroughly evaluate the non-standard copula model used in this work.*
>
> We appreciate this comment and we take this very seriously.  We working to complete two major new experiments in order to more thoroughly evaluate the copula model that we present.  The first involves an analysis of the optimal tail parameter of the marginal distribution, which will help us to better characterize the observed shape of the marginals.   The second involves a comparison of our method versus standard copula (Gaussian, Archimedian), as well as a comparison against expressive high dimensional probability distributions including characteristic functions.  We discuss this proposed analysis in greater detail in our response to all reviewers.
>
>
> **Weakness 2.**  *The evaluation of the distribution over activations in CNNs is only empirical and thus limited in its applicability.*
>
> We have carefully constructed our new experiment (major revision 1) to expand our empirical analysis of the marginals to include an analysis of the optimal Weibull tail parameter $\theta$ and a discussion of how this tail parameter compares with the theoretical analysis of Vladimirova (2019) suggesting an optimal Weibull tail parameter of $0.5 i$ where $i$ is the layer depth.   We are very curious as to extent that the theoretical result of Vladimirova (2019) using Bayesian CNNs and Gaussian inputs may exhibit supporting empirical evidence in the more realistic setting of popular CNNs and image datasets which is the experimental backdrop of our paper.
>
>
> **Weakness 3.**  *The paper lacks a thorough discussion of related work and a precise formal definition of the method itself.*
>
> (3.1 related work)
> We are working to greatly expand the related literature section, including the papers suggested by yourself and related papers in the areas of long-tailed feature distributions, copula analysis, high dimensional probability density, and out of distribution detection (as a downstream application).  More discussion of the related work we will include is in our response to all reviewers.
>
> We would also like to thank you very much for your paper suggestions, they were very helpful!
>
>
> (3.2 formal definition)
> Regarding precise formal definition, we would like to clarify with you that you are asking about the definition of the experimental design [p7+] as opposed to the definitions within the methodology [p3-6].
>
> The reason why we are asking is because in your detailed remarks you ask:  "[p.7-9] How exactly the copula is defined is unclear. What exactly is the formal definition used in this work? Also, how is it estimated? Only on page 10 is an informal definition given."
>
> To address this, we will be including two appendices, to provide more detailed information regarding the exact experimental design and additional definitions to be able to reproduce the experiments.  These appendices will define the sampling methods employed, hypothesis test performed, evaluation criteria (i.e. which version of KL-divergence), the version of the probability integral transform used, and other relevant definitions toward reproducibility.
>
> **Weakness 4.**  *The results are very limited and, even though potentially interesting to some, might be insufficient for publication at TMLR.*
>
> We take this feedback very seriously, and we are working expand our experimental analysis toward two major revisions with new analysis.
>
>    1.  Empirical estimation of the optimal Weibull tail parameter $\theta$   (est. 1 week)
>    2.  Comparison of Copula Inter-dependence versus related methods   (est. 3 weeks)
>
>  We discuss these new experiments in greater detail in our overall response to reviewers.  Experiment 1 will help us to put our analysis results in the context of Long-tailed theory, and Experiment 2 will help us to put our analysis in the context of related high-dimensional probability density methods.
>
>
> *Detailed Changes:*
>
>    We would also like to thank you for spotting several detailed minor revisions.  We will address these point by point in our updated submission (est. 1 week)

---

> > ### Comment · Reviewer_FLh4 · 2025-03-31
> >
> > Thank you for the response. I look forward to seeing the updated manuscript.
> >
> > Regarding (3.2 formal definition): When reading section 2 (methodology), it should be clear what exactly the proposed methodology is. Currently, this is (in my opinion) missing. This could mean introducing a final subsection in which the method is explained as a whole or (ideally) formalised in a concise way. This way, it would be clear how the previous subsections connect and what the proposed method looks like.

---

### Author Response · Authors · 2025-03-31
**Response to All Reviewers**

We greatly appreciate all of the constructive feedback that we have received regarding this paper.
We are working to implement several major revisions, including substantial new analysis.  We believe that these revisions will address many of the key concerns as discussed by the constructive feedback.

Due to the extent of this new analysis, we would like to formally ask the reviewers as well as the action editor for some additional time.

The first proposed major revision: analysis of Weibull tail parameter, is essentially completed.  We will have an updated manuscript within one week to present our expanded results and updated finding, as well as a number of minor revisions that we also describe here.

Beyond this we will require time to complete our second proposed major revision: expanded analysis of the copula interdependence.  In this section we intend to add additional comparison with existing techniques from the literature.  We estimate including this as a separate revision within three weeks.

We have below a more detailed discussion of the proposed major and minor revisions,

Proposed Major Revisions (substantial new analysis):

1.  Empirical estimation of the optimal Weibull tail parameter $\theta$   (est. 1 week)

   The reviewers raise a major concern that 'exponential view' hypothesis is inadequately justified, and is not discussed in the context of related work on 'long-tailed' feature distributions.  Notably the work of Feldman et al. suggests the presence of long-tailed deep features are necessary memorize rare examples. Vladimirova proves that Baysian networks (including Bayesian CNNs) with Gaussian training data exhibit long-tailed deep features, with optimal Weibull tail parameter $\theta=0.5i$ where $i$ is the depth of the layer.

   We propose to carefully analyze the Weibull tail parameter $\theta$ which is necessary to characterize if the distribution is actually long-tailed, exponential, or sub-exponential.  In particular, we are very intrigued whether the theoretical result of Vladimirova of $\theta=0.5i$ for Bayesian CNNs with Gaussian data and will continue to show empirical evidence for the popular CNN architectures and image datasets that we analyze in this paper.  We have implemented a new experiment to perform this comparison and are very excited to be including these results in the near future into an updated version of the paper.

2.  Comparison of Copula Inter-dependence versus related methods   (est. 3 weeks)

   The reviewers bring out a very important point that there are other techniques for estimating the probability density of high dimensional distributions.  Notably Characteristic functions and characteristic circuits are considered to be advanced techniques.  Moreover, our paper needs to make more clear what is the advantage of our novel non-parametric copula design versus standard copula designs such as the Gaussian and Archimedean copulas.

   We are working to expand the analysis of the fitness of the Copula Inter-dependence to include a comparison with these existing techniques, in order to provide a more comprehensive discussion of how we model the high dimensional deep feature distribution, in a way that is both expressive (in terms of modeling complex relations), as well as easy to visualize with separated marginal and interdependence plots.



Proposed Minor Revisions  (additional writing, but no new analysis):

1.  Expanded Literature Discussion   (est.  1 week)

	Citation and discussion of Long-tailed feature distributions

	Citation and discussion of Orthogonal polynomials in feature representation

	Citation and discussion of high-dimensional density functions

	Citation and discussion of applications of Copula and potential relevance

	In-depth discussion of Out-of-Distribution (OOD) detection as a downstream task including discussion of deep feature density based methods.

2.  Clarification of experimental design  (est. 1 week)

	Appendix A.  detailed experimental design of marginal analysis, including discussion of objective criteria, sampling methods, hypothesis testing method, and hyperparameter settings.

	Appendix B.  detailed experimental design of copula interdependence, including definition of the KL-divergence criteria used, as well as sampling methods, and hyperparameter settings.

3.  Additional minor revisions as suggested by reviewers


Thank you all of your feedback, we really appreciate it!  And we ask your patience as we work to finalize these revisions.  With kind regards.

---

### Author Response · Authors · 2025-06-05
**Completion of Major Revisions**

Dear reviewers,

   We would like to thank you all for your constructive feedback regarding this manuscript.  We have completed substantial major revisions to address the criticisms and concerns that you have identified.  These major revisions include:

1.  A greatly expanded the literature review, with a discussion of long-tailed theory, and related density methods.

2.  A new formal definition plus algorithmic pseudocode to clarify the methodology.

3.  A new experimental analysis to determine the extent to which deep feature marginals are long-tailed.

4.  A new comparison of deep feature tail parameters with theoretical estimates.

5.  A new experimental intercomparison with related methods: Archimedean copulas and ECF.

6.  New detailed appendices to further clarify all experimental design details and evaluation criteria.

   We believe that this revision is a substantial improvement to the manuscript, both in terms of the academic rigor as well as the clarity of the presentation.  We greatly appreciate your constructive feedback that has made this major revision possible, and look forward to your review of this revised work.

---

### Decision · Action_Editor_75oV · 2025-06-25

**Recommendation:** Reject

**Additional Comments:**

This paper considers the problem of estimating the PDF of the features of a CNN.  This problem is generally difficult due to curse of dimensionality. Authors propose an empirical technique that combines copula analysis with method of moments.


This paper was reviewed by three expert reviewers and received the following recommendations: Leaning Accept, Leaning Reject, Reject. I think paper is studying an interesting topic but authors are not able to convince the reviewers sufficiently well about the merits of their results. The following concerns were brought up by the reviewers:

- The reviewer who is the most critical did not find authors' response adequate. In particular,  the reivewer raises the concerns: 1- novelty of core hypothesis (though not a main issue for decision) 2- limited discussion on practicality 3- relation to prior works/methods. The reviewer was not satisfied with authors' response on the raised concerns/issues.

- The issue that the motivation is not clear was raised by another reviewer. Ultimately, the reviewer thinks that the modeling the density of activations is not a task we encounter frequently, and the authors have not demonstrated sufficiently well why the reader should care about this task. Although I think the problem is interesting, I agree that the problem needs better motivation for readers who are not in this research area.

- Reviewers generally found that the results are too limited, with little prractical motivation. Authors' response seems to not convince them sufficiently well on this issue.


No reviewers championed the paper, they are not particularly excited about the paper.
Two reviewers recommended rejecting the paper, one of which is a strong reject.
As such, based on the reviewers' suggestion,  I recommend not including this version of the paper to the TMLR.

**Audience:**

No

**Audience Explanation:**

AE estimates that the audience for this paper would be limited, based on the reviewer comments. Specifically, two reviewers have the following comments:
1- "I am still struggling to see why this paper might be of interest to the TMLR community."
2- "This work might target a too specific audience and the results are potentially too limited."

The third reviewer was also not satisfied with the (lack of) connections to practical applications.

**Claims And Evidence:**

Yes

**Claims Explanation:**

The claims made in this paper is mainly empirical and supported by numerical studies.

**Resubmission Of Major Revision:**

The authors may consider submitting a major revision at a later time.